# Systems level identification of a matrisome-associated macrophage polarisation state in multi-organ fibrosis

John F Ouyang[1,2]*[†], Kunal Mishra[1,2][†], Yi Xie[1,2], Harry Park[1,2], Kevin Y Huang[1,2], Enrico Petretto[1,2,3]*, Jacques Behmoaras[1,2,4]*

[1]Centre for Computational Biology, Duke-NUS Medical School, Singapore, Singapore; [2]Programme in Cardiovascular and Metabolic Disorders, Duke-NUS Medical School, Singapore, Singapore; [3]Institute for Big Data and Artificial Intelligence in Medicine, School of Science, China Pharmaceutical University (CPU), Nanjing, China; [4]Department of Immunology and Inflammation, Centre for Inflammatory Disease, Imperial College London, London, United Kingdom

**\*For correspondence:**
john.ouyang@duke-nus.edu.sg (JFO);
enrico.petretto@duke-nus.edu.sg (EP);
jacquesb@duke-nus.edu.sg (JB)

[†]These authors contributed equally to this work

**Competing interest:** The authors declare that no competing interests exist.

**Abstract** Tissue fibrosis affects multiple organs and involves a master-regulatory role of macrophages which respond to an initial inflammatory insult common in all forms of fibrosis. The recently unravelled multi-organ heterogeneity of macrophages in healthy and fibrotic human disease suggests that macrophages expressing osteopontin (SPP1) associate with lung and liver fibrosis. However, the conservation of this SPP1[+] macrophage population across different tissues and its specificity to fibrotic diseases with different etiologies remain unclear. Integrating 15 single-cell RNA-sequencing datasets to profile 235,930 tissue macrophages from healthy and fibrotic heart, lung, liver, kidney, skin, and endometrium, we extended the association of SPP1[+] macrophages with fibrosis to all these tissues. We also identified a subpopulation expressing matrisome-associated genes (e.g., matrix metalloproteinases and their tissue inhibitors), functionally enriched for ECM remodelling and cell metabolism, representative of a matrisome-associated macrophage (MAM) polarisation state within SPP1[+] macrophages. Importantly, the MAM polarisation state follows a differentiation trajectory from SPP1[+] macrophages and is associated with a core set of regulon activity. SPP1[+] macrophages without the MAM polarisation state (SPP1[+]MAM[-]) show a positive association with ageing lung in mice and humans. These results suggest an advanced and conserved polarisation state of SPP1[+] macrophages in fibrotic tissues resulting from prolonged inflammatory cues within each tissue microenvironment.

## Editor's evaluation

This important study deepens our understanding of macrophage phenotypes in pathological contexts and identifies a new macrophage state associated with tissue fibrosis, as well as putative drivers of this cellular state. The authors provide convincing evidence and performed a well-thought-out and thoroughly described computational analysis of single-cell RNA sequencing data. This work will be of broad interest to the fields of tissue inflammation, fibrosis, macrophage biology, and immunology.

## Introduction

Fibrosing diseases comprise a multitude of human organs and share a common end-point: unresolved inflammation characterised by abnormal production of extracellular matrix (ECM) and

interstitial scar formation. In almost all tissues, macrophages control the pathobiology of fibrosis in a timely manner. Macrophage polarisation accompanies the progressively changing tissue microenvironment and macrophages modulate fibroblast activation and ECM-producing myofibroblast *trans*-differentiation through soluble factors or direct cell-cell interaction (*Novak and Koh, 2013*; *Setten et al., 2022*; *Wynn and Barron, 2010*; *Wynn and Vannella, 2016*). The outcome of fibrosis is directly dependent on tissue macrophage heterogeneity as different macrophage subpopulations can have contrasting modulatory effects on fibrogenesis (*Duffield et al., 2005*). For instance, in the lung and heart, monocyte-derived macrophages infiltrate the inflamed tissues and show pro-fibrogenic activity (*Chen et al., 2022*; *Misharin et al., 2017*). On the contrary, tissue resident macrophages negatively regulate fibrosis as their loss exacerbates cardiac and lung fibrosis (*Chakarov et al., 2019*; *Zaman et al., 2021*).

Although fibrotic lesions may occur in distinct anatomical sites within the same organ, the process of fibrogenesis is a shared hallmark of several diseases, including heart failure, chronic kidney disease (CKD), liver cirrhosis, and interstitial lung disease (*Zeisberg and Kalluri, 2013*). In addition, fibrosis of the skin and uterus are seen in patients diagnosed with systemic sclerosis (SSC) (*Denton and Khanna, 2017*) and endometriosis (*Nishimoto-Kakiuchi et al., 2023*; *Vigano et al., 2018*; *Viganò et al., 2020*), respectively. All these conditions share an inflammatory component with monocyte/macrophage involvement in affected organs. Thus, knowledge of homeostatic (disease-free) tissue macrophage heterogeneity is required to understand the role of macrophage subpopulations and their polarisation during fibrosis in these etiologically different diseases. To this aim, cross-tissue human single-cell atlas initiatives provided a useful resource to address the complexity of homeostatic tissue resident macrophage states at an unprecedented resolution in multiple organs (*Domínguez Conde et al., 2022*; *Eraslan et al., 2022*; *McEvoy et al., 2022*; *Sikkema et al., 2023*; *Jones et al., 2022*). In addition to these resources, recent single-cell transcriptomics studies compared healthy and fibrotic (or inflammatory) human tissues, and further revealed the evolution of the macrophage polarisation states during inflammation/fibrosis in multiple organs (*Adams et al., 2020*; *Ayaub et al., 2021*; *Gur et al., 2022*; *Koenig et al., 2022*; *Kuppe et al., 2021*; *Lake et al., 2023*; *Malone et al., 2020*; *Morse et al., 2019*; *Ramachandran et al., 2019*; *Rao et al., 2021*; *Reyfman et al., 2019*; *Valenzi et al., 2019*).

The findings that have emerged from these large-scale, single cell-based resources using healthy and fibrotic human tissues can be summarised in three points: (1) the heterogeneity of macrophages is conserved across human tissues with tissue-restricted functionalisation (e.g., expression of genes related to iron recycling in erythrophagocytic macrophages of spleen and liver Kupffer cells *Domínguez Conde et al., 2022*); (2) a macrophage transcriptome broadly associates with lipid-related pathways during homeostasis in multiple human tissues (*Eraslan et al., 2022*); (3) among other cell markers, a disease-associated macrophage population express the matricellular glycoprotein osteopontin (*SPP1*), a gene implicated in the development of wound healing and fibrosis (*Liaw et al., 1998*; *Mori et al., 2008*; *Pinto, 2021*; *Rotem et al., 2022*). Importantly, SPP1+ macrophages (also termed as scar-associated macrophages [SAMs], initially described in hepatic fibrosis [*Fallowfield et al., 2007*] and further refined at a single-cell level in cirrhotic liver [*Ramachandran et al., 2019*]) have been described as pro-fibrotic cells in human pulmonary and hepatic fibrosis (*Morse et al., 2019*; *Ramachandran et al., 2019*; *Reyfman et al., 2019*; *Fabre et al., 2023*). However, the broader implication of SPP1+ SAMs in multi-organ fibrosis and their potential heterogeneity remain to be identified. Furthermore, SPP1+ SAMs share an overlapping transcriptome with TREM2+ tumor-associated macrophages (TAMs) (*Mulder et al., 2021*), lipid-associated macrophages (LAMs) (*Jaitin et al., 2019*), and disease-associated microglia (DAM) (*Keren-Shaul et al., 2017*; *Silvin et al., 2022*) – hence lack specific functionality for tissue fibrosis.

Here, we focused on human tissues and analysed single-cell RNA-sequencing (scRNA-seq) data from 15 studies carried out in healthy and disease tissues characterised by fibrosis of the heart, lung, liver, kidney, skin, and uterus. Focusing on the tissue macrophage compartment (235,930 cells), we show that SPP1+ macrophages are indeed a prominent feature conserved in multi-organ fibrosis in humans. Stemming from SPP1 macrophages, we identified a functionally defined polarisation state, called matrisome-associated macrophage (MAM), which we investigated further.

## Results

### SPP1⁺ macrophages increase during fibrotic disease across tissues in humans

Based on the previously established association of pro-fibrotic SPP1⁺ macrophages with cirrhotic liver and lungs from idiopathic pulmonary fibrosis (IPF) patients (*Morse et al., 2019*; *Ramachandran et al., 2019*), we hypothesised that this macrophage population can be detected in other human tissues and can associate with broader fibrotic disease state. Thus, we interrogated human single-cell datasets which contain, among other immune cells, macrophages in healthy and fibrotic tissues. For consistency and ease of normalisation for downstream meta-analysis, we focused on datasets generated in a single platform (10X), using live cell isolation protocols (*Table 1*). This led to a total of 235,930 tissue monocyte/macrophages from healthy and diseased human liver, lung, heart, skin, endometrium, and kidney (*Table 1*). The pathologies affecting these organs included cirrhosis, nonalcoholic steatohepatitis (NASH), IPF, SSC, ischemic cardiomyopathy (ICM), dilated cardiomyopathy (DCM), keloid scarring, endometriosis, CKD, and acute kidney injury (AKI). The percentage of macrophages vary across the different datasets, ranging from 44.3% of total cells in the lung (*Adams et al., 2020*), to 0.7% in skin (*Deng et al., 2021*).

We then performed data-driven clustering of the monocyte/macrophages within each tissue (see Materials and methods, *Figure 1—figure supplements 1–6*, *Figure 1—source data 1*). We obtained between 5 and 17 clusters across the six human tissues and calculated marker genes for these clusters, which we then overlaid with marker genes for other macrophage states such as TAMs, LAMs, DAM, and SAMs, as well as tissue-specific macrophages such as FABP4⁺ alveolar macrophages and MARCO⁺ liver macrophages (Kupffer cells; *Figure 1—figure supplements 1–6*, *Figure 1—source data 2*). Across all six tissues, SPP1⁺ macrophages are identified as a common cell population (*Figure 1A–F*). When compared with all macrophages, the SPP1⁺ macrophages are positively enriched for several processes, including ECM degradation/remodelling and metabolic processes such as oxidative phosphorylation (*Figure 1—source data 3*). Furthermore, when compared with healthy control tissue, the proportions of SPP1⁺ macrophages were consistently increased in cirrhotic/NASH liver (*Figure 1A*), IPF/SSC lung (*Figure 1B*), ICM/DCM heart (*Figure 1C*), keloid/SSC skin (*Figure 1D*), endometriosis uterus (*Figure 1E*), and CKD/AKI kidney (*Figure 1F*). When analysed at an individual (control/patient) level, the increase in the proportions of SPP1⁺ macrophages with disease was recapitulated in all tissues, with the exception of endometrium, though liver, skin, and heart did not reach statistical significance (*Figure 1G*). Amongst the top marker genes for SPP1⁺ macrophages, *SPP1* itself was consistently upregulated across six tissues together with *GPNMB*, *CAPG*, and *ALCAM* (*Figure 1H*). These results confirmed and extended the association of SPP1⁺ macrophages with multi-organ fibrosis, and further prioritised additional genes that can be used to distinguish this macrophage subset.

### A MAM state of polarisation within SPP1⁺ cells

We next reasoned that SPP1⁺ macrophages may be heterogeneous and contain polarisation state(s) that can be used for a refined functional characterisation of these cells during multi-organ fibrosis. Single-cell transcriptomics capture a continuum of macrophage phenotypes within patient tissues who often show differences in disease severity. As such, the transcriptional resolution of a cell state in a single tissue can be confounded by several factors such as sample size, sequencing depth, disease heterogeneity, etc. We thus performed an integrative analysis of SPP1⁺ macrophages across six tissues, including 10 fibrotic conditions and their matched controls. Using unsupervised clustering (*Figure 2—figure supplement 1A–D*), we identified a MAM subcluster (defined as SPP1⁺MAM⁺) which had further increased *SPP1* expression amongst SPP1⁺ macrophages (*Figure 2—figure supplement 1E*, *Figure 2—source data 1*). The remaining macrophages (defined as SPP1⁺MAM⁻) were enriched for inflammatory processes (*Figure 2—figure supplement 1F*). The tissue origin of the macrophages was not a bias in the subcluster identification within the SPP1⁺ cells (*Figure 2—figure supplement 1A*). For clarity of nomenclature, we refer to SPP1⁺ macrophages as those defined in multiple tissues during organ fibrosis (*Figure 1* and *Figure 1—figure supplements 1–6*), to SPP1⁺MAM⁺ macrophages as the cells derived from subclustering of SPP1⁺ macrophages in the single-cell meta-analysis (*Figure 2A* and *Figure 2—figure supplement 1*), and to SPP1⁺MAM⁻ macrophages as the SPP1⁺ population devoid of MAM⁺ cells.

**Table 1.** Summary of scRNA-seq datasets analysed in this study.

For each study, the sample size (number of single-cell datasets), age/gender, disease status, the number (and percentage) of tissue resident macrophages (Mφ), and the total number of live cells sequenced are indicated. For studies that opted for positive or negative cell sorting methods (e.g., CD45[+] cells), the total number of live cells is indicated with the cell isolation method. IPF: idiopathic pulmonary fibrosis, SSC: systemic sclerosis; DCM: dilated cardiomyopathy; ICM: ischemic cardiomyopathy; CKD: chronic kidney disease; AKI: acute kidney injury. [1]The number of CD45[+] cells has been derived from the dataset downloaded from GEO (otherwise this number has been provided in the original publication and reported here). [2]Massive parallel single-cell RNA-sequencing (Mars-seq) was used to profile single-cell transcriptomics instead of 10X Chromium kit.

| Tissue | Study (ref in article) | Repository | Disease status | Sample size (n) | Sex (%M) | Age (years)± SD | Tissue Mφ (%) | Total cells (#) |
|---|---|---|---|---|---|---|---|---|
| Liver | *Ramachandran et al., 2019* | GSE136103 | Cirrhosis | 5 | 80.0 | 56.6±5.8 | 8332 (13.9%) | 60,094 CD45[+] cells[1] |
| | | | Normal | 5 | 60.0 | 57.4±7.9 | | |
| Liver | *Fred et al., 2022* | Author provided data | NASH | 10 | 50.0 | 47.0±7.5 | 2069 (12.1%) | 17,154 |
| Lung | *Morse et al., 2019* | GSE128033 | IPF | 3 | 33.3 | 69.3±0.6 | 17,570 (39.3%) | 44,652 |
| | | | Normal | 4 | 50.0 | 38.3±20.6 | | |
| Lung | *Reyfman et al., 2019* | GSE122960 | IPF | 4 | 75 | 66.5±5.0 | 32,136 (41.7%) | 77,079 |
| | | | Normal | 8 | 25 | 42.9±15.5 | | |
| | | | SSC | 2 | 0 | 46.0±9.9 | | |
| Lung | *Adams et al., 2020* | GSE136831 | IPF | 32 | 81.3 | 65.4±5.4 | 117,184 (37.4%) | 312,928 |
| | | | Normal | 17 | 58.8 | 44.4±18.9 | | |
| Lung | *Valenzi et al., 2019* | GSE128169 GSE156310 | IPF | 1 | 100.0 | 68.0 | 17,407 (44.3%) | 39,252 |
| | | | SSC | 4 | 75.0 | 56.8±9.5 | | |
| Heart | *Koenig et al., 2022* | GSE183852 | DCM | 5 | 60.0 | 50.0±19.2 | 3,922 (7.9%) | 49,665 |
| | | | Normal | 2 | 100.0 | 50.5±17.7 | | |
| Heart | *Rao et al., 2021* | GSE145154 | DCM | 2 | 100.0 | 61.0±1.4 | 20,539 (30.0%) | 68,516 CD45[+] cells |
| | | | ICM | 4 | 100.0 | 47.3±10.6 | | |
| | | | Normal | 1 | 100.0 | 53.0 | | |
| Skin[2] | *Gur et al., 2022* | GSE195452 | Normal | 22 | 22.7 | 44.8±10.4 | 2456 (15.6%) | 15,700 CD45[+] cells |
| | | | SSC | 55 | 5.4 | 50.1±13.2 | | |
| Skin | *Deng et al., 2021* | GSE163973 | Keloid | 3 | 66.7 | 25.6±7.3 | 320 (0.7%) | 45,094 |
| | | | Normal | 3 | 66.7 | 31.0±7.0 | | |
| Endo- metrium | *Tan et al., 2022* | GSE179640 | Endometriosis | 9 | 0.0 | 35.8±5.8 | 3079 (8.6%) | 35,941 |
| | | | Normal | 3 | 0.0 | 33.3±10.3 | | |
| Endo- metrium | *Fonseca et al., 2023* | GSE213216 | Endometriosis | 22 | 0.0 | 34.1±7.6 | 2951 (1.8%) | 163,882 |
| | | | Normal | 8 | 0.0 | 37.9±10.1 | | |
| Kidney | *Kuppe et al., 2021* | Zenodo 4059315 | CKD | 6 | 66.7 | 70.8±12.5 | 3596 (7.5%) | 48,096 CD10[-] cells |
| | | | Normal | 4 | 100.0 | 65.3±11.3 | | |
| Kidney | *Lake et al., 2023* | atlas.kpmp.org | AKI | 11 | 72.7 | 50.2±19.1 | 1597 (3.5%) | 46,249 |
| | | | CKD | 15 | 66.7 | 61.9±12.7 | | |
| | | | Normal | 14 | 50.0 | 45.9±10.3 | | |
| Kidney | *Malone et al., 2020* | GSE145927 | AKI | 3 | 100.0 | 49.3±18.2 | 2772 (4.6%) | 60,080 |
| | | | Normal | 2 | 50.0 | 54.0±1.0 | | |

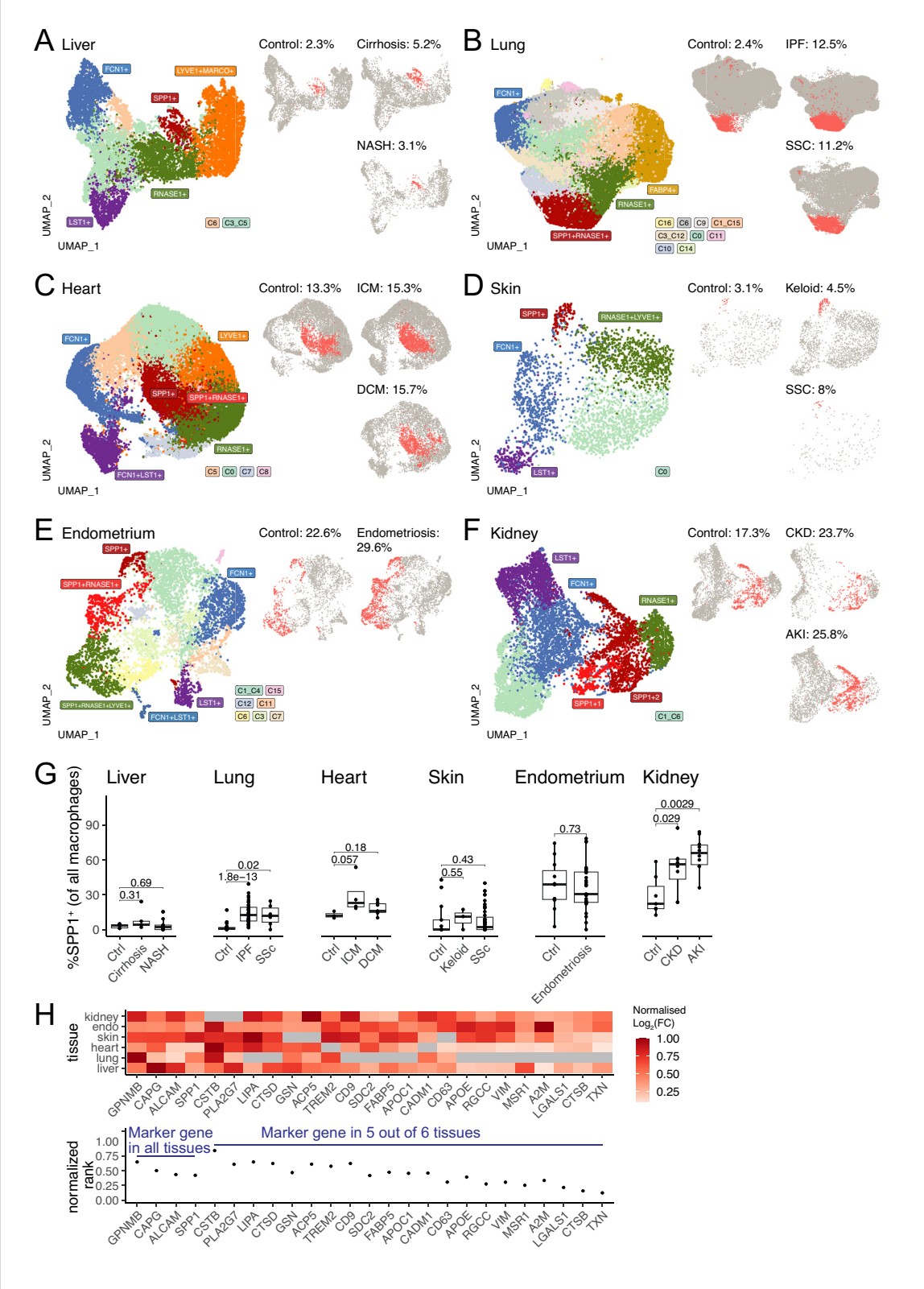

**Figure 1.** SPP1+ macrophages are increased during fibrotic disease across human tissues. (**A–F**) Uniform Manifold Approximation and Projection (UMAP) dimensionality reduction of all monocytes and macrophages in single-cell RNA-sequencing (scRNA-seq) of liver (**A**), lung (**B**), heart (**C**), skin (**D**), endometrium (**E**), and kidney (**F**) from fibrotic disease patients or controls (see *Table 1* for details on cell and sample numbers). Specific clusters were denoted as SPP1+ macrophages, RNASE1+/LYVE1+ homeostatic macrophages, MARCO+/FABP4+ tissue-specific macrophages, FCN1+ classical

*Figure 1 continued on next page*

*Figure 1 continued*

monocytes, or LST1⁺ non-classical monocytes based on expression of marker genes (see *Figure 1—figure supplements 1–6* for details), while the remaining transitional macrophages were denoted based on their cluster ID from unsupervised clustering. SPP1⁺ macrophages in each tissue are stratified by disease conditions and coloured in red on separate UMAP; the proportion of SPP1⁺ macrophages out of all macrophages in each condition is indicated. (**G**) Boxplot summarising the relative proportion of SPP1⁺ macrophages (of all macrophages) in each subject stratified by disease status. Liver: (control) n=5, (cirrhosis) n=5, (nonalcoholic steatohepatitis [NASH]) n=9; lung: (control) n=29, (idiopathic pulmonary fibrosis [IPF]) n=40, (systemic sclerosis [SSC]) n=6; heart: (control) n=3, (ischemic cardiomyopathy [ICM]) n=4, (dilated cardiomyopathy [DCM]) n=7; skin: (control) n=13, (keloid) n=3, (SSC) n=39; endometrium: (control) n=11, (endometriosis) n=27; kidney: (control) n=7, (chronic kidney disease [CKD]) n=8, (acute kidney injury [AKI]) n=10. The Wilcoxon rank-sum test was used to evaluate the significance of the difference between groups. (**H**) Heatmap displaying the log₂-fold-change (FC) of the top differentially expressed genes (DEGs) (x-axis) upregulated in SPP1⁺ macrophages compared to other macrophages across different tissues (top panel). For each gene, the log₂FC was scaled to the highest value in each tissue and non-DEGs are denoted in grey. Rank-plot prioritising top DEGs conserved across tissues (bottom panel), where the x-axis shows marker genes and y-axis represents the cumulative rank of each DEG based on log₂FC within each tissue.

The online version of this article includes the following source data and figure supplement(s) for figure 1:

**Source data 1.** Marker genes for monocyte / macrophages.

**Source data 2.** Marker genes for TREM2+ macrophages.

**Source data 3.** GSEA analysis of DEG between SPP1+ macrophages and other macrophages in each tissue.

**Figure supplement 1.** Data-driven clustering of liver macrophages.

**Figure supplement 2.** Data-driven clustering of lung macrophages.

**Figure supplement 3.** Data-driven clustering of heart macrophages.

**Figure supplement 4.** Data-driven clustering of skin macrophages.

**Figure supplement 5.** Data-driven clustering of endometrium macrophages.

**Figure supplement 6.** Data-driven clustering of kidney macrophages.

We defined an SPP1⁺MAM⁺ transcriptomic signature comprising 158 genes (*Figure 2—source data 2*). In order to identify markers that delineate SPP1⁺MAM⁺ polarisation state in an unbiased way, we used COMET (*Delaney et al., 2019*; *Figure 2—figure supplement 2* and see Materials and methods) and prioritised *CBLB, MITF, RASAL2, GPC4, CD52*, and *RGCC* (*Figure 2B*). Because of the potential transcriptional overlap between the SPP1⁺MAM⁺ and previously defined tissue macrophage states such as LAMs (*Jaitin et al., 2019*), TAMs (*Mulder et al., 2021*), DAM (*Keren-Shaul et al., 2017*; *Silvin et al., 2022*), and SAMs (*Ramachandran et al., 2019*), we next performed a systematic comparative analysis across these cell states (*Figure 2C* and see *Figure 2—figure supplements 3 and 4* for derivation of DAM and LAM cells, respectively). This analysis revealed, among others, *CHI3L1, MMP9, GPC4* as genes uniquely differentially expressed and upregulated in the SPP1⁺MAM⁺ polarisation state (*Figure 2C*). As expected, some markers (e.g., *SPP1, LPL,* and *MATK*) were shared across LAMs, DAM, and TAMs, and as such, did not add context specificity to the SPP1⁺MAM⁺ state (*Figure 2C*).

To achieve further transcriptional specificity, we reasoned that the activation of matrisome-related genes polarises the SPP1⁺ macrophages further, and that without the MAM signature, the cells will recap the transcriptional state of the previously identified SAMs (*Ramachandran et al., 2019*; *Figure 2—figure supplement 4A*). Indeed, we found that hepatic SPP1⁺MAM⁺ macrophages form a subset of SAMs (*Figure 2—figure supplement 4B*). In keeping with this, the markers of SAMs are differentially expressed in the SPP1⁺MAM⁺ macrophages (*Figure 2—figure supplement 4C–D*). When compared with SPP1⁺MAM⁻, gene set enrichment analysis (GSEA) of differentially expressed genes indicated predominance of ECM remodelling and cell metabolism-related pathways, including 'osteoclast development' in SPP1⁺MAM⁺ (*Figure 2D*, *Figure 2—source data 3*). Interestingly, when we queried a proteomics-based matrisome database (*Naba et al., 2012*), pathway enrichment for SPP1⁺MAM⁺ and SPP1⁺MAM⁻ differentially expressed genes was distinct. SPP1⁺MAM⁺ showed an association with ECM regulators, while SPP1⁺MAM⁻ state was enriched for secreted factors (*Figure 2E*). Specifically, SPP1⁺MAM⁺ macrophages are characterised by expression of matrix metalloproteinases and their tissue inhibitors (*MMP7, MMP9, MMP19, TIMP3*) and cathepsin family genes with endopeptidase activity (*CTSK*), suggesting these cells contribute to ECM remodelling that occur in multiple tissues during fibrosis. On the other hand, SPP1⁺MAM⁻ are enriched specifically for secreted factors and cytokine-cytokine receptor signalling (*Figure 2E*), as well as for interferon/complement and immune-related processes (*Figure 2—figure supplement 1F* and *Figure 2—source data 3*). These results

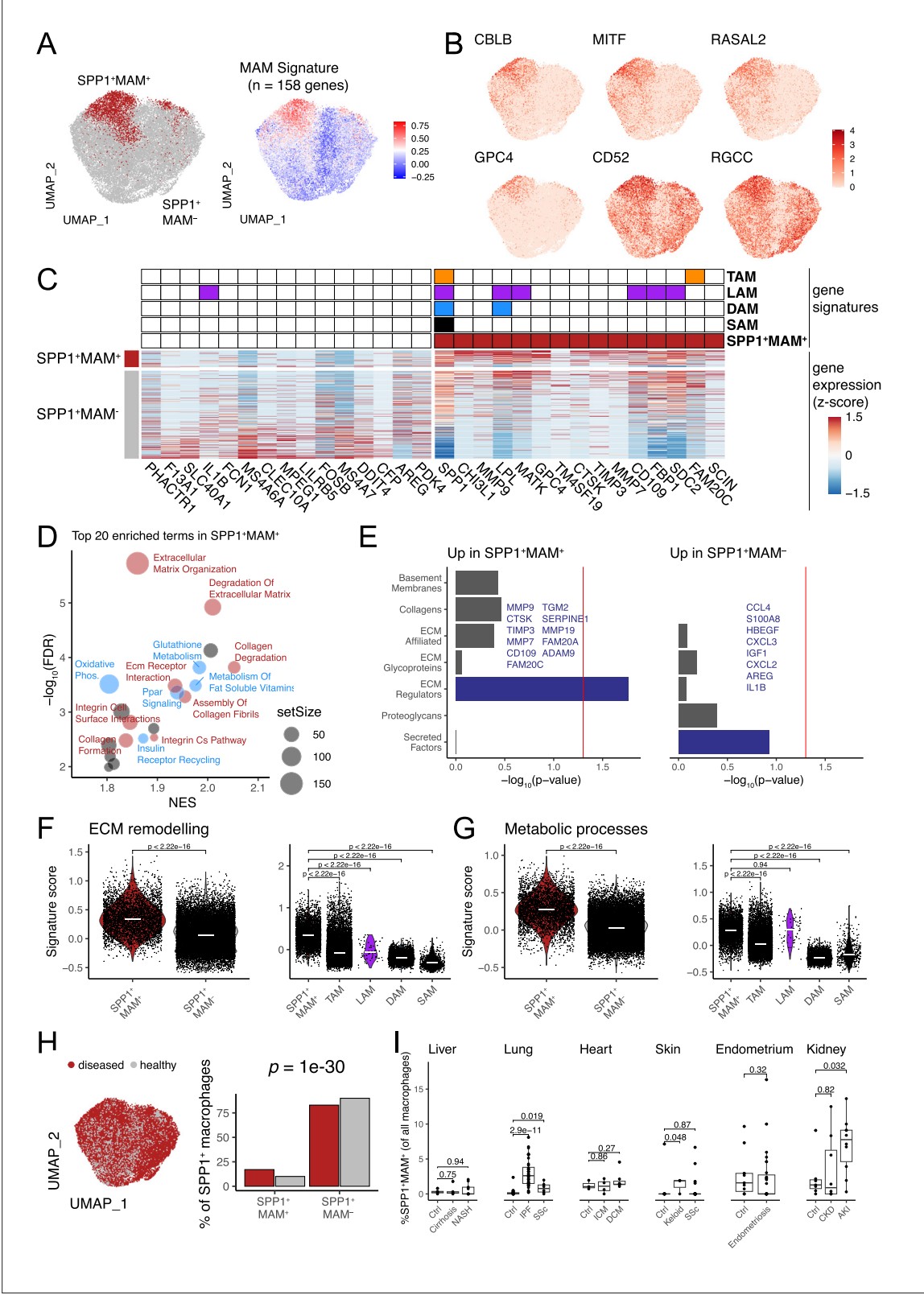

**Figure 2.** Identification of a matrisome-associated macrophage (MAM) state within SPP1+ macrophages. (**A**) Uniform Manifold Approximation and Projection (UMAP) dimensionality reduction of all SPP1+ macrophages merged from the liver, lung, heart, skin, endometrium, and kidney tissues following data integration (see Materials and methods). Unsupervised clustering of SPP1+ macrophages identified a MAM subpopulation (red), denoted as SPP1+MAM+ (left panel; *Figure 2—figure supplement 1*). Each macrophage is coloured by the expression level of SPP1+MAM+ signature genes

*Figure 2 continued*

(n=158 genes; right panel). (**B**) UMAP of SPP1+ macrophages with each cell coloured by the expression of top markers for SPP1+MAM+ macrophages as predicted by COMET (see Materials and methods and *Figure 2—figure supplement 2*). (**C**) Heatmap showing the single-cell gene expression levels of top 15 differentially expressed genes (DEGs), based on log$_2$-fold-change, between SPP1+MAM+ and SPP1+MAM- macrophages. Markers of SPP1+MAM+, tumour-associated macrophages (TAMs) (*Mulder et al., 2021*), lipid-associated macrophages (LAMs) (*Jaitin et al., 2019*), disease-associated microglia (DAM) (*Keren-Shaul et al., 2017*; *Silvin et al., 2022*), and scar-associated macrophages (SAMs) (*Ramachandran et al., 2019*) are indicated in the heatmap. (**D**) Scatterplot summarising the results of gene set enrichment analysis (GSEA) of differential expression between SPP1+MAM+ and SPP1+MAM- macrophages. For each significant pathway (FDR <0.05), x-axis indicates the normalised enrichment score (NES) and y-axis the significance of the enrichment; the size of each dot is proportional to the number of genes in the gene set. Metabolism-related pathways are coloured in blue, extracellular matrix (ECM) remodelling in red, and remaining terms in grey. (**E**) Bar plots summarising pathway enrichment of DEG upregulated in SPP1+MAM+ (left panel) or in SPP1+MAM- macrophages (right panel) using a matrisome-specific database (*Naba et al., 2012*). p-Values are calculated using the hypergeometric test and red line indicates a p-value of 0.05. Overlapping genes in the top-enriched pathways are displayed. (**F, G**) Violin plots showing the transcriptomic signature scores (y-axis) of SPP1+MAM+ ECM remodelling gene set (n=18 genes) (**F**), and SPP1+MAM+ metabolic processes gene set (n=27 genes) (**G**) in SPP1+MAM+ and SPP1+MAM- macrophages (left), and in other macrophage subsets with overlapping transcriptomic signatures (right). SPP1+MAM+, n=3840; SPP1+MAM-, n=20,265; TAMs, n=16,130; LAMs, n=61; DAM, n=5645; SAMs, n=415 cells. (**H**) UMAP dimensionality reduction of SPP1+ macrophages across all tissues, coloured by the disease status (left panel). Proportion of SPP1+MAM+ or SPP1+MAM- amongst SPP1+ macrophages from disease or healthy samples. The over-representation of SPP1+MAM+ cells in disease samples was evaluated by hypergeometric test and indicated. (**I**) Boxplot showing the percentage of SPP1+MAM+ across all macrophages, stratified by tissue and disease/healthy status. Lung: (control) n=29, (idiopathic pulmonary fibrosis [IPF]) n=40, (SSC) n=6; heart: (control) n=3, (ischemic cardiomyopathy [ICM]) n=4, (dilated cardiomyopathy [DCM]) n=7; skin: (control) n=13, (keloid) n=3, (SSC) n=39; endometrium: (control) n=11, (endometriosis) n=27; kidney: (control) n=7, (chronic kidney disease [CKD]) n=8, (acute kidney injury [AKI]) n=10. Unless otherwise indicated, a two-sided Wilcoxon rank-sum test was used to evaluate the significance of differences between groups.

The online version of this article includes the following source data and figure supplement(s) for figure 2:

**Source data 1.** Marker genes for subclusters within SPP1+ macrophages.

**Source data 2.** Signature genes of SPP1+MAM+ macrophages.

**Source data 3.** GSEA analysis of pathways upregulated/downregulated in SPP1+MAM+ macrophages (relative to SPP1+MAM- macrophages).

**Source data 4.** Signature genes of "ECM-remodeling processes" upregulated in SPP1+MAM+ macrophages.

**Source data 5.** Signature genes of "metabolic processes" upregulated in SPP1+MAM+ macrophages.

**Figure supplement 1.** Data-driven clustering of SPP1+ macrophages identifies a matrisome-associated macrophage polarisation state.

**Figure supplement 2.** Markers of SPP1+MAM+ as predicted by COMET.

**Figure supplement 3.** Derivation of disease-associated microglia (DAM) transcriptomic signature from the temporal lobe of epilepsy patients.

**Figure supplement 4.** Derivation of scar-associated macrophages (SAMs) from human liver data.

suggest a phenotypically distinct state of polarisation of SPP1+MAM+ macrophages. Since the 'ECM remodelling' and 'metabolic processes' predominantly define SPP1+MAM+ state, we interrogated the presence of these two pathway signatures (*Figure 2—source data 4–5*) within the previously reported macrophage states such as TAM, LAM, DAM, SAM (*Figure 2F and G*). SPP1+MAM+ macrophages showed significant upregulation of ECM remodelling and metabolic processes-related gene signatures compared to SPP1+MAM- macrophages (*Figure 2F and G*). Notably, when compared with TAMs, DAM LAMs, and SAMs, SPP1+MAM+ showed increased ECM remodelling (*Figure 2F*). The same is true for the metabolic processes, with the exception of LAMs (*Figure 2G*), which are specialised in metabolic homeostasis (*Jaitin et al., 2019*). Overall, our gene signature analysis confirmed a distinct transcriptome profile of SPP1+MAM+ macrophages that may implicate a polarisation state associated with ECM remodelling and metabolic reprogramming during multi-organ fibrosis. The SPP1+MAM+ macrophages were present in healthy and disease tissues with a significant increase in proportion during disease state (*Figure 2H*). At an individual (control/patient) level, SPP1+MAM+ macrophages were significantly increased in IPF lung, SSC lung, keloid skin, and AKI kidney (*Figure 2I*).

## SPP1+MAM- macrophages polarise towards SPP1+MAM+ state during human fibrotic disease

Having established the distinct transcriptional and functional signatures of SPP1+MAM+ cells, we next investigated their relationship with the other macrophage subpopulations and their differentiation state via single-cell trajectory analysis. Slingshot differentiation trajectory analyses identified a main trajectory from the FCN1+ infiltrating monocyte/macrophages towards the homeostatic (RNASE1+) macrophages and another one from FCN1+ monocyte/macrophages towards SPP1+MAM+

macrophages (*Figure 3A–F*). We repeated the analysis with Monocle (*Figure 3—figure supplement 1*) and validated similar trajectories towards homeostatic and SPP1⁺MAM⁺ macrophages. Importantly, SPP1⁺MAM⁺ cells were at the end of the trajectory and preceded by the SPP1⁺MAM⁻ state (*Figure 3A–F*) in all tissues except the kidney. These results suggest that SPP1⁺MAM⁺ macrophages may represent a conserved terminal polarisation state arising from SPP1⁺ macrophages.

When stratified according to the disease status, the cell propensity score (CPS, the probability of SPP1⁺MAM⁻ macrophages to differentiate into SPP1⁺MAM⁺ ones) was significantly more prominent in at least one fibrotic disease state throughout all six tissues (*Figure 3G*). This also means that, compared with the SPP1⁺MAM⁻ macrophages in controls, there are significantly more SPP1⁺MAM⁻ macrophages with high propensity to differentiate into SPP1⁺MAM⁺ in each tissue where fibrosis can occur. Interestingly, some healthy tissues, such as the liver, endometrium, and kidney had a relatively elevated CPS compared with healthy lung, heart, and skin, which showed negligible propensity of differentiation of SPP1⁺MAM⁻ towards SPP1⁺MAM⁺. This may be partly due to the control liver samples being obtained from non-lesional tissues of solitary colorectal cancer patients. Similarly, control endometrium samples originated from patients with eutopic endometrium. Thus, for these two tissues, the increased baseline propensity may reflect a not entirely healthy state.

## Regulons that associate with SPP1⁺MAM⁺ polarisation state in humans

Since SPP1⁺MAM⁺ cells are at the end trajectory of different macrophage states, and are likely to be a hallmark of multi-organ fibrosis, we next investigated transcription factor (TF) networks (regulons) that might be involved in their differentiation. The SPP1⁺MAM⁺ state of polarisation was associated with the likelihood of activation of several regulons (*Figure 4A*). Amongst these, JDP2, EGR1, MEF2C, ELK3, KLF3, and CEBPD regulons showed the highest specificity for the SPP1⁺MAM⁺ signature (*Figure 4B*). We next investigated the association between SPP1⁺MAM⁺ and the activity of each regulon during the differentiation trajectories from transitional (i.e., macrophages that are not homeostatic/RNASE1⁺ nor SPP1⁺), SPP1⁺MAM⁻ and SPP1⁺MAM⁺ states. To this end, we fit a linear regression model for the top 3 TFs in *Figure 4A* (i.e., JDP2, KLF3, and CEBPD) to assess the SPP1⁺MAM⁺ regulon association along the two differentiation trajectories: (1) from transitional towards SPP1⁺MAM⁻ and (2) from SPP1⁺MAM⁻ to SPP1⁺MAM⁺ macrophages (*Figure 4C*). Both differentiation trajectories showed similar correlations between the regulon activity score and the SPP1⁺MAM⁺ signature score, suggesting that these regulons are active in cells prior their polarisation into SPP1⁺ macrophages. Furthermore, the activities of these regulons were positively associated with the SPP1⁺MAM⁺ polarisation state when all SPP1⁺ macrophages were considered (*Figure 4D*, *Figure 4—source data 1*). The regulon activity of the top 3 TFs in SPP1⁺MAM⁺ macrophages was increased with respect to homeostatic/RNASE1⁺ macrophages (*Figure 4E*). Taken together, these results prioritise JDP2, KLF3, and CEBPD regulon activities that associate with the differentiation towards the SPP1⁺MAM⁺ polarisation state in multiple human tissues.

## SPP1⁺MAM⁻ polarisation state is associated with aging in healthy human and mice lung tissues

Physiological ageing is one of the risk factors for fibrotic disease (*Kapetanaki et al., 2013*; *Rehan et al., 2021*) which is often associated with impaired resolution of the prior inflammatory insults (*De Maeyer et al., 2020*) and with metabolically activated macrophages (*Minhas et al., 2021*; *Minhas et al., 2019*). Capitalising on the Human Lung Cell Atlas (*Sikkema et al., 2023*), we next evaluated signatures of homeostatic, SPP1⁺MAM⁻, SPP1⁺MAM⁺, ECM remodelling (SPP1⁺MAM⁺), metabolic processes (SPP1⁺MAM⁺) as a function of age in healthy individuals' lung tissues (*Figure 5A*, *Figure 5—source data 1–6*). We found a significant positive correlation between SPP1⁺MAM⁻ activity score and age (p=0.001), and to a lesser extent, between homeostatic activity score and age (p=0.011). However, neither SPP1⁺MAM⁺ signature nor ECM processes derived from SPP1⁺MAM⁺ macrophages were associated significantly with older age (*Figure 5A and B*). It should be noted that significance for the homeostatic activity score is also dependent on the smoking status of the individuals (*Figure 5—source data 2*). Indeed, stratifying the individuals according to their smoking status resulted in loss of significant association between the macrophage homeostatic activity score and age (*Figure 5—figure supplement 1*). We extended this analysis by adding healthy murine lung macrophages from the Tabula Muris Senis atlas (*Almanzar et al., 2020*; *Figure 5C* and *Figure 5—figure supplement 2*),

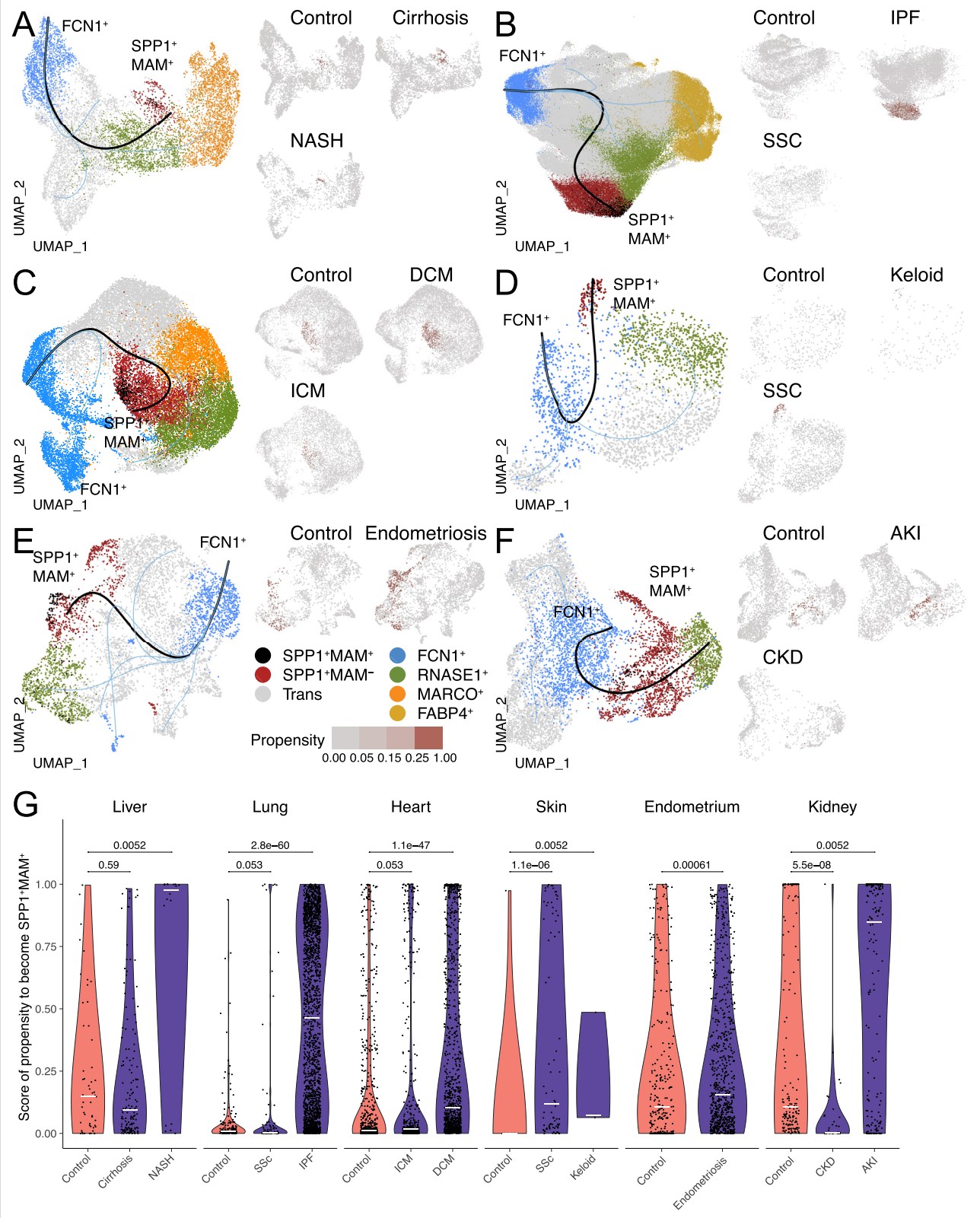

**Figure 3.** The differentiation trajectory of SPP1⁺MAM⁺ macrophages across human tissues. (**A–F**) Slingshot differentiation trajectory analyses of macrophages in the liver (**A**), lung (**B**), heart (**C**), skin (**D**), endometrium (**E**), and kidney (**F**). The predicted trajectories are drawn on Uniform Manifold Approximation and Projection (UMAP) projection of macrophages, where the trajectory from FCN1⁺ monocytes to SPP1⁺MAM⁺ macrophages is indicated by a black line, and the one from FCN1⁺ monocytes to other clusters is indicated by blue lines. Macrophages are further stratified by disease

*Figure 3 continued*

and controls in each tissue and plotted separately on different UMAP, with each SPP1$^+$MAM$^-$ macrophage coloured by its propensity to differentiate towards the SPP1$^+$MAM$^+$ state. To improve the visualisation of the propensity score of each cell across all tissues, we binned the score into four bins with different shades of blue colour. (**G**) Violin plots displaying the SPP1$^+$MAM$^-$ macrophage propensity score towards the SPP1$^+$MAM$^+$ state in each tissue, separated by disease conditions. Number of cells is as follows, liver: (control) n=53, (cirrhosis) n=140, (nonalcoholic steatohepatitis [NASH]) n=18; lung: (control) n=167, (SSC) n=78, (idiopathic pulmonary fibrosis [IPF]) n=2,006; heart: (control) n=662, (ischemic cardiomyopathy [ICM]) n=421, (dilated cardiomyopathy [DCM]) n=1001; skin: (control) n=16, (keloid) n=3, (systemic sclerosis [SSC]) n=78; endometrium: (control) n=296, (endometriosis) n=735; kidney: (control) n=213, (chronic kidney disease [CKD]) n=40, (acute kidney injury [AKI]) n=110. Colour of violin delineates disease (purple) and control (orange). Unless otherwise indicated, a two-sided Wilcoxon rank-sum test was used to evaluate the statistical significance (p-value) of differences between two groups.

The online version of this article includes the following figure supplement(s) for figure 3:

**Figure supplement 1.** Monocle analysis recapitulates the conserved differentiation trajectory from FCN1$^+$ monocytes to SPP1$^+$ macrophages across multiple tissues.

which supported an increased SPP1$^+$MAM$^-$ activity score with ageing in mouse. Overall, these analyses suggest that there is increased SPP1$^+$MAM$^-$ polarisation with age, which may further progress into a disease-associated SPP1$^+$MAM$^+$ state when tissue fibrosis occurs.

## Discussion

A cell polarisation state has been broadly characterised by macrophages expressing lipid metabolism-associated genes in different organs. Depending on the tissue or disease context, these macrophages have been described as LAMs (obesity, *Jaitin et al., 2019*) or LAM-like (homeostasis, *Eraslan et al., 2022*) or TAMs (cancer, *Mulder et al., 2021*) or DAM (Alzheimer's disease, *Keren-Shaul et al., 2017*; *Silvin et al., 2022*) or lipid-droplet-accumulating microglia (ageing brain *Marschallinger et al., 2020*). Inarguably many genes that characterise these macrophage polarisation states show context specificity; nonetheless, certain genes have been repeatedly found throughout different studies and they include *TREM2*, *FABP5*, genes belonging to cholesterol metabolism (*APOE*, *APOC1*, *LPL*) and cell adhesion (*SPP1*, *CD9*). This ubiquitous macrophage transcriptional state seems to be conserved across tissues and is seen during homeostasis and disease (*Domínguez Conde et al., 2022*; *Eraslan et al., 2022*; *McEvoy et al., 2022*; *Sikkema et al., 2023*; *Jones et al., 2022*; *Koenig et al., 2022*; *Lake et al., 2023*; *Morse et al., 2019*; *Ramachandran et al., 2019*; *Reyfman et al., 2019*; *Fabre et al., 2023*; *Mulder et al., 2021*; *Jaitin et al., 2019*; *Keren-Shaul et al., 2017*; *Silvin et al., 2022*; *Remmerie et al., 2020*). During fibrotic pathologies, SAMs also display lipid-associated transcriptional markers and although they associate with hepatic and pulmonary fibrotic disease (*Morse et al., 2019*; *Ramachandran et al., 2019*; *Remmerie et al., 2020*), their phenotypic difference with regard to the ubiquitous LAM state remains incompletely understood. In this study, we focus on SPP1$^+$ macrophages and extend their known association with lung and liver fibrosis to other tissues such as the endometrium. Interestingly, fibrosis is an increasingly recognised feature of endometriosis (*Vigano et al., 2018*; *Guo, 2018*). Given the recent cell atlas of endometrium revealing disease-associated macrophages (*Tan et al., 2022*), our results implicate a potential role of SPP1$^+$ macrophages during endometrial fibrosis.

Here, we propose osteopontin (*SPP1*) as a suitable marker for pro-fibrotic macrophages in multiple tissues. *TREM2* is another candidate marker gene that has been proposed for such association. In addition to studies performed in humans (*Ramachandran et al., 2019*), single cell-based transcriptome analysis in murine models of human fibrotic disease has confirmed the existence of a Trem2-positive macrophage population associated with disease (*Remmerie et al., 2020*; *Joshi et al., 2020*; *Jung et al., 2022*; *Seidman et al., 2020*; *Subramanian et al., 2021*; *Xie et al., 2018*; *Xiong et al., 2019*), even though some studies suggest a pro-resolution role of Trem2 itself (*Jung et al., 2022*; *Perugorria et al., 2019*) or of Trem2$^+$ macrophages (*Daemen et al., 2021*). Furthermore, regenerative Trem2$^+$ macrophages mitigate fibrosis after skin transplantation in humans (*Henn et al., 2021*). These studies suggest that Trem2 is not an optimal marker for SAMs, and in line with the proposed overall pro-regenerative role of Trem2$^+$ LAMs (*Guilliams and Scott, 2022*), Trem2$^+$ macrophages may represent a hybrid state with features of both resident macrophages and infiltrating monocytes during tissue regeneration (*Ramachandran et al., 2019*). SPP1, on the other hand, is a biomarker of fibrosis

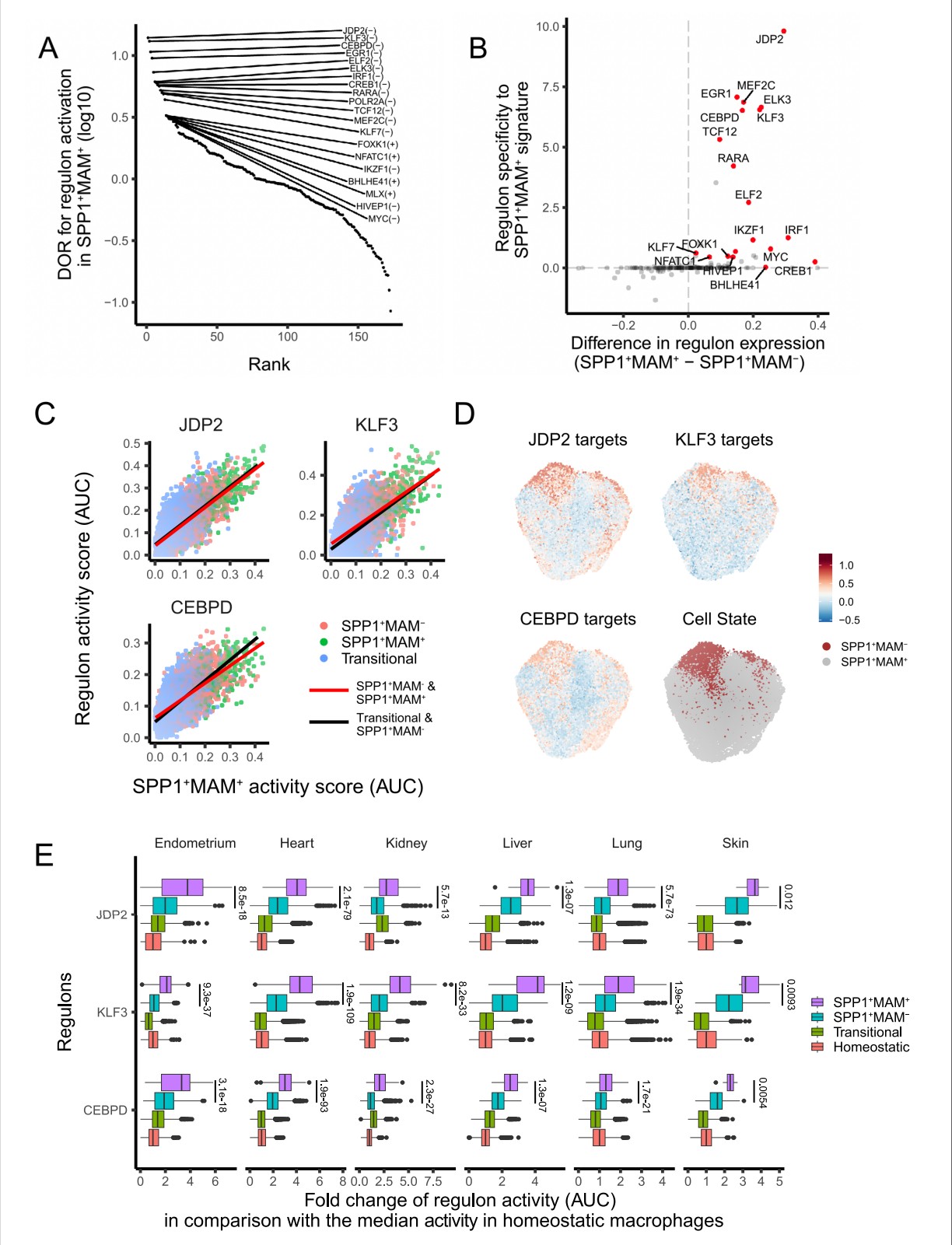

**Figure 4.** A core set of regulon activity associate with SPP1⁺MAM⁺ differentiation. (**A**) Rank plot for the 173 regulons active in SPP1⁺MAM⁺ macrophages, ordered by the diagnostic odd ratio (DOR) (y-axis), which calculates the odds of the regulon being activated in SPP1⁺MAM⁺ cells over the odds of being activated in SPP1⁺MAM⁻ macrophages (see Materials and methods). A positive DOR is associated with the regulon activation in SPP1⁺MAM⁻ macrophages. (**B**) Scatterplot showing the specificity of regulon genes to the SPP1⁺MAM⁺ signature genes (y-axis) against the difference in expression

*Figure 4 continued on next page*

*Figure 4 continued*

of a regulon between SPP1⁺MAM⁺ and SPP1⁺MAM⁻ macrophages (x-axis). Briefly, the specificity scores the degree of overlap between regulon genes and the SPP1⁺MAM⁺ signature genes (see Materials and methods). (**C**) Scatterplot showing each cell's activity score of selected regulons (y-axis) against the cell's SPP1⁺MAM⁺ signature activity score (x-axis) across all tissues. Cells are coloured based on their identity: transitional, SPP1⁺MAM⁻ and SPP1⁺MAM⁺ macrophages. For each regulon, separate linear regression models have been fitted for each trajectory: from the transitional to SPP1⁺MAM⁻ macrophages (black regression line), and from SPP1⁺MAM⁻ to SPP1⁺MAM⁺ macrophages (red regression line). (**D**) Uniform Manifold Approximation and Projection (UMAP) of SPP1⁺ macrophages coloured by the expression level of selected regulons, and by SPP1⁺MAM⁺ or SPP1⁺MAM⁻ status. (**E**) Boxplots summarising the fold-change (FC) of regulon gene expression from SPP1⁺MAM⁺, SPP1⁺MAM⁻, transitional macrophages compared with regulon expression in homeostatic macrophages. Within each tissue, the ratio (FC) between the regulon activity score (area under the curve score [AUC]) in each cell from SPP1⁺MAM⁺, SPP1⁺MAM⁻, or transitional macrophage with respect to the median AUC in the homeostatic macrophages was calculated. The differences in FCs in regulon expression between SPP1⁺MAM⁺ macrophages and other macrophages (i.e., SPP1⁺MAM⁺ vs SPP1⁺MAM⁻, SPP1⁺MAM⁺ vs transitional macrophages, or SPP1⁺MAM⁺ vs homeostatic macrophages) were all statistically significant (adjusted p-value <0.05) after a two-tailed Wilcoxon rank-sum test. Adjusted p-value for comparisons between SPP1⁺MAM⁻ and SPP1⁺MAM⁺ macrophages are indicated.

The online version of this article includes the following source data for figure 4:

**Source data 1.** Member genes of regulons listed in *Figure 4D*.

in NASH (*Glass et al., 2018*), primary sclerosing cholangitis (*De Muynck et al., 2023*), and of interstitial lung disease progression in SSC (*Gao et al., 2020*). Unlike Trem2, there is unequivocal evidence for the pro-fibrotic role of SPP1 during multi-organ fibrosis. Importantly, Spp1 deletion or neutralisation in mice attenuates fibrosis in different models of kidney, heart, lung, liver, skin, prostate, and muscle injuries (*Coombes et al., 2016*; *Dong and Ma, 2017*; *Honda et al., 2020*; *Kiefer et al., 2010*; *Kumar et al., 2022a*; *Leung et al., 2013*; *Matsui et al., 2004*; *Persy et al., 2003*; *Popovics et al., 2021*; *Vetrone et al., 2009*; *Wu et al., 2012*; *Yoo et al., 2006*). SPP1 is a constituent of ECM but can also be found as a secreted soluble factor. In ECM-rich fibrotic tissues, SPP1 can mediate the kinetics of the attachment of the macrophages to the extracellular microenvironment.

Despite being a marker of SAMs, *SPP1* has also been described as a marker of the generalised LAM state. Lipid-related pathways, and in particular, fatty acid oxidation (FAO) in non-myeloid cells has been previously linked to renal fibrosis (*Kang et al., 2015*) and targeting macrophage FAO holds therapeutic potential in pulmonary fibrosis (*Gu et al., 2019*). In order to uncover potential pathways other than lipid-related ones, here we provide a detailed single-cell map of the transcriptional heterogeneity of SPP1⁺ macrophages and further identify a MAM polarisation state within multiple human tissues. This state is characterised by the (1) upregulation of transcripts belonging to ECM remodelling such as MMPs, TIMPs, and cathepsin family genes, (2) metabolic processes, (3) activation of regulons including those involved in osteoclast development (JDP2, NFATC1). Interestingly, SPP1⁺MAM⁻ macrophages are enriched for secretory and inflammatory/immune pathways, suggesting that MAM polarisation state is acquired from phenotypically inflammatory SPP1⁺ macrophages. Indeed, the trajectory analysis showed that SPP1⁺MAM⁺ polarisation state followed the SPP1⁺MAM⁻ in multiple human tissues, further arguing that SPP1⁺macrophages, independently of tissue microenvironment, show a common differentiation path. Future work is required to optimise the tissue sorting and in situ characterisation of SPP1⁺MAM⁺ macrophages based on markers specifically induced in this cell state (*CBLB*, *GPC4*, *CD52*). This can also allow a thorough metabolic profiling of SPP1⁺MAM⁺ macrophages, which can provide mechanistic insights beyond the lipid-related pathways that broadly characterise SPP1⁺ macrophages. As per the MAM nomenclature, we refer to this as a state of polarisation rather than a macrophage subpopulation and make systematic distinction between SPP1⁺MAM⁻ and SPP1⁺MAM⁺. As pertinently argued (*Guilliams and Scott, 2022*), and in accordance with findings presented here, SPP1 expressing LAMs are found in steady state, thus nomenclature based on activation trajectories of these cells (rather than pathology-related states) could be more appropriate.

Our results suggest a transcriptional resemblance between SPP1⁺MAM⁺ macrophages and osteoclasts. Evidence for the latter is supported by expression of osteoclast genes (*MMP9*, *CTSK*) and/or osteoclast network genes (*Kang et al., 2014*; *Pereira et al., 2020*) by MAM⁺ cells, the significant association of MAM state with osteoclast development, and regulons such as JDP2 and NFATC1 that regulate osteoclast differentiation (*Asagiri and Takayanagi, 2007*; *Kawaida et al., 2003*; *Maruyama et al., 2012*). The osteoclast resemblance of SPP1⁺MAM⁺ macrophages may also explain their significant enrichment for pathways related to ECM remodelling and energy metabolism. Multinucleated osteoclasts depend on mitochondrial oxidative phosphorylation for their resorptive activity

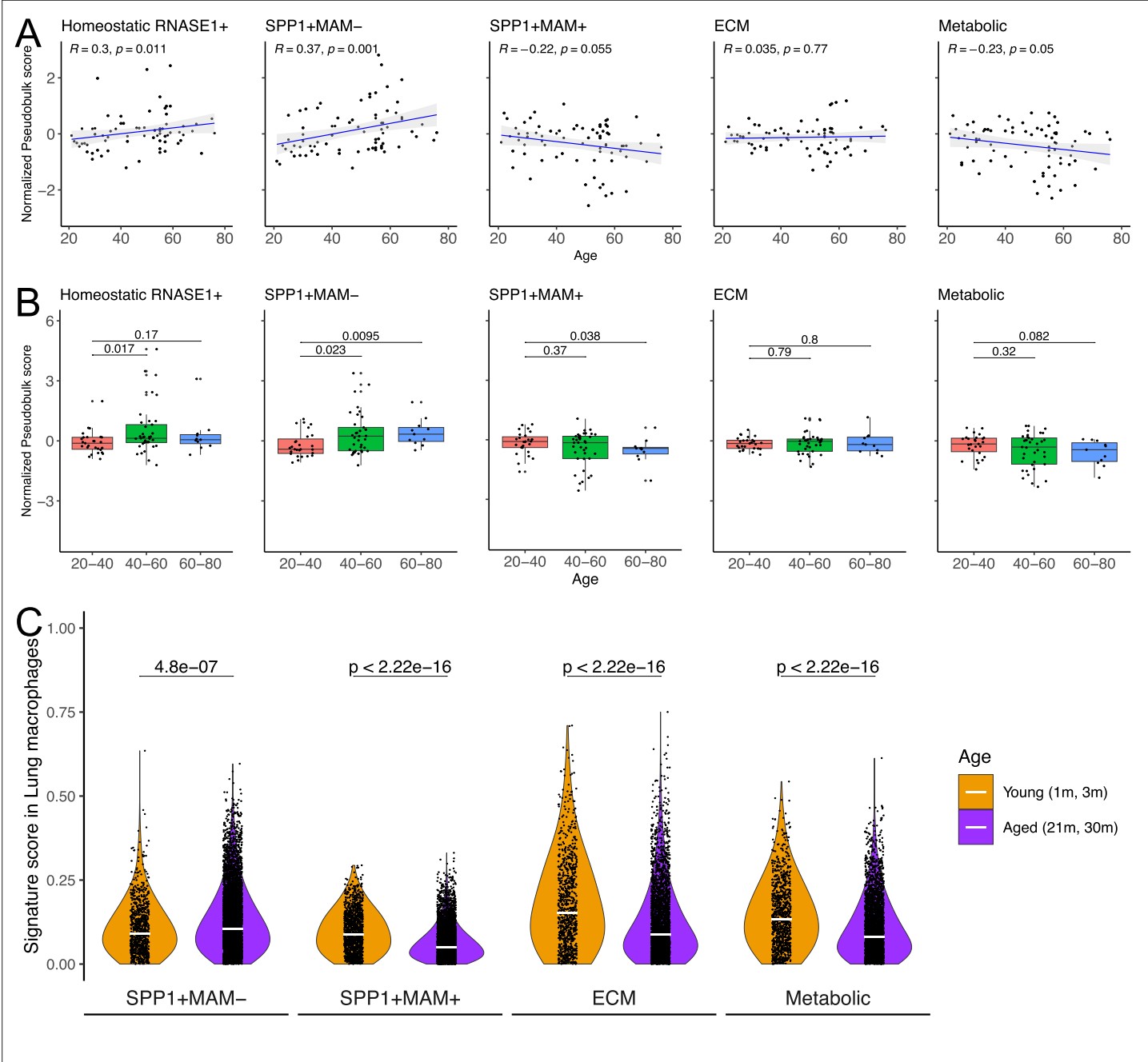

**Figure 5.** SPP1+MAM- gene signature is associated with ageing in mice and humans. (**A**) Scatterplot evaluating the pseudo-bulk expression of the homeostatic RNASE1+, SPP1+MAM-, SPP1+MAM+, SPP1+MAM+ extracellular matrix (ECM) remodelling, and SPP1+MAM+ metabolic processes signatures (y-axis) in healthy human lung macrophages against age (x-axis) Here, for each sample, the pseudo-bulk expression signature was z-scaled after taking the median of the signature score from each cell. For each regression, the grey band around the line represents the 95% confidence interval of the regression line. Number of individuals = 85, number of cells = 86,484. (**B**) Boxplot summarising the pseudo-bulk expression of the homeostatic RNASE1+, SPP1+MAM-, SPP1+MAM+, SPP1+MAM+ ECM remodelling, and SPP1+MAM+ metabolic processes signatures in healthy human lung macrophages, stratified by age groups. (**C**) Violin plots summarising the SPP1+MAM-, SPP1+MAM+, SPP1+MAM+ ECM remodelling, and SPP1+MAM+ metabolic processes signatures in healthy mouse lung macrophages taken from Tabula Muris Senis (*Almanzar et al., 2020*), stratified into aged (21–30 months) or young mice (1–3 months). Aged, n=7878 cells, young, n=1529 cells. Unless otherwise indicated, two-tailed Wilcoxon rank-sum test was used to evaluate the statistical significance (p-value) of differences between the groups.

The online version of this article includes the following source data and figure supplement(s) for figure 5:

**Source data 1.** Signature genes of SPP1+MAM- macrophages.

*Figure 5 continued on next page*

*Figure 5 continued*

**Source data 2.** Homeostatic signature regression model.

**Source data 3.** SPP1+MAM- signature regression model.

**Source data 4.** SPP1+MAM+ signature regression model.

**Source data 5.** SPP1+MAM+ ECM signature regression model.

**Source data 6.** SPP1+MAM+ Metabolic signature regression model.

**Figure supplement 1.** Homeostatic macrophage signatures modelled against age and stratified by smoking status.

**Figure supplement 2.** SPP1$^+$MAM$^-$ signature is associated with ageing in mice kidneys.

that remodels the bone matrix (*Jin et al., 2014*; *Lemma et al., 2016*). Interestingly, Trem2$^+$ LAMs expressing Spp1 showed a transcriptional profile similar to osteoclasts during advanced stages of murine atherosclerosis (*Cochain et al., 2018*), and human multinucleated giant cells express MAM markers such as SPP1, MMP9, CHI3L1 in granulomatous slack skin (*Feng et al., 2023*). Multinucleated adipose tissue macrophages, probably derived from SPP1-expressing LAMs, also show functional and morphological resemblance with osteoclasts (*Olona et al., 2021*). As the name indicates, osteopontin was first cloned in the bone (*Franzén and Heinegård, 1985*; *Oldberg et al., 1986*) and has subsequently been proposed to mediate the adhesion of osteoclasts to resorbing bone (*McKee et al., 2011*; *Reinholt et al., 1990*).

One outstanding question remains: are SPP1$^+$MAM$^+$ macrophages pro-fibrotic? Association with fibrotic disease does not necessarily imply polarization towards a disease-promoting state. Besides, given the degradative enzymes that define SPP1$^+$MAM$^+$ macrophages (MMPs and cathepsins) and their overlap with osteoclast-related transcriptional pathways, it could be argued that these cells differentiate from SPP1$^+$ macrophages to promote healing by counteracting prominent tissue fibrosis. We thus postulate that (1) SPP1$^+$ macrophages are involved in the initial states of fibrosis and show pro-fibrotic activity across multiple tissues through a secretory phenotype and through activation of fibroblasts, (2) the MAM transcriptional signature is acquired as a compensatory phenotype to counteract established fibrosis at later stages. In addition to topological localisation of SPP1$^+$ macrophages to ECM-producing fibroblasts (*Bhattacharya and Ramachandran, 2023*), evidence for (1) is manifold and growing. Through secreted factors, SPP1$^+$ macrophages devoid of MAM signature can facilitate macrophage-fibroblast interaction and cooperation in fibrogenesis (*Buechler et al., 2021*). Specifically, CXCL4 has been proposed to be a critical cytokine for myofibroblast activation that is promoted by SPP1$^+$ macrophages (*Hoeft et al., 2023*). Furthermore, GM-CSF, IL-17A, and TGF-β1 can promote differentiation of SPP1$^+$ macrophages, which contribute to collagen I-associated fibrosis (*Fabre et al., 2023*). Although SPP1 has been described to be expressed by non-myeloid cells and to regulate the production of ECM components in the lung and liver (*Pardo et al., 2005*; *Urtasun et al., 2012*), macrophage-derived SPP1 can induce migration and proliferation of fibroblasts (*Gao et al., 2020*). A recent study also predicted preferential interaction between SPP1$^+$ TAMs and activated fibroblasts to promote pro-tumorigenic ECM (*Zhang et al., 2020*). A second line of evidence for (1) is macrophage phenotype during physiological ageing, a condition that is associated with low-grade chronic inflammation that possibly triggers early fibrogenesis (*Adler et al., 2020*). Inflammaging (defined as unresolved systemic inflammation in the absence of pathogens; *Franceschi et al., 2000*; *Franceschi et al., 2018*) is characterised by tissue macrophages becoming progressively pro-inflammatory (*van Beek et al., 2019*) and ageing alveolar macrophages do express SPP1 (*Angelidis et al., 2019*), suggesting that chronic unresolved inflammation causes early fibrogenesis in tissues. In keeping with this, we found significant association of SPP1$^+$MAM$^-$ (but not MAM$^+$) macrophages with ageing in the lung. Conversely, during fibrotic pathology where tissues may show more advanced fibrosis, SPP1$^+$MAM$^-$ macrophages show increased propensity to polarise towards the MAM$^+$ state, a matrix-degrading and osteoclast-like phenotype. Hence, SPP1$^+$ macrophages may show dynamic states of polarisation depending on the degree of inflammation/fibrosis in the tissue microenvironment. The exact function of SPP1$^+$MAM$^+$ macrophages, including their in-depth phenotypic characterisation in the fibrotic niche, is to be experimentally validated and comparatively studied with regard to SPP1$^+$MAM$^-$ macrophages – one of the limitations of this study.

A converging body of evidence supports the infiltrating monocyte origin of pro-fibrotic tissue macrophages (*Misharin et al., 2017*; *Sikkema et al., 2023*; *Remmerie et al., 2020*; *Seidman et al.,*

*2020*; *Daemen et al., 2021*; *Hulsmans et al., 2018*; *Rizzo et al., 2023*). Our single-cell trajectory analyses extend this observation to six different human tissues, adding further resolution to the SPP1+ macrophages. According to the continuum model of monocyte-to-macrophage differentiation (or activation paths, *Sanin et al., 2022*), a healthy tissue environment is maintained by the ability of the monocyte-derived macrophages to undergo homeostatic differentiation into tissue resident macrophages (*Park et al., 2022*). The disease state is accompanied by monocyte-derived macrophages infiltrating tissues, and progressively losing the ability to support tissue resident macrophages because of the pathologically evolving tissue-specific environmental cues (*Park et al., 2022*). For instance, SPP1+ macrophage recruitment to the fatty liver from circulating monocytes coincides with the lack of Kupffer cells (*Remmerie et al., 2020*). Hence, the fine balance between tissue repair and fibrosis could be explained by these spatiotemporally regulated infiltrating monocytes and their differentiation state within the tissue microenvironment (*Park et al., 2022*). Our results fit with this model and we propose that the MAM+ state is an advanced polarisation state of monocyte-derived macrophages that show conserved activation pathways due to prolonged inflammatory and fibrogenic cues in multiple human tissues. A detailed understanding of MAM state across different tissues can refine anti-fibrotic treatment initiatives.

## Materials and methods

### Derivation of monocyte/macrophage clusters from each dataset

Raw or pre-filtered UMI count matrices for the datasets used in this study were downloaded either from GEO repositories or zenodo repositories or atlas websites or obtained directly from authors (see *Table 1* for more details). The R programming language (v4.2.2) and Seurat package (v4.3) (*Stuart et al., 2019*) was used to perform subsequent quality control filtering and single-cell analysis. Quality control to remove low-quality cells was first applied to all 15 datasets prior to integrated analysis. For each dataset, cells expressing between 300 and 5000 genes, less than 20% mitochondrial reads (genes starting with '*MT-*'), and less than 0.1% haemoglobin reads (*HBA*, *HBB*) were retained. For one of the skin datasets (*Gur et al., 2022*), a more lenient cutoff of less than 40% mitochondrial reads with the expressed genes/haemoglobin read cutoff described above was applied as cells in this dataset show relatively higher proportions of mitochondrial reads. Upon filtering, transcripts expressed in one or more cell were retained for downstream analyses.

To identify the monocyte/macrophages from each dataset, a two-step filtering is applied to each dataset (identification followed by removal of non-monocyte/macrophage cells). For each set of cell filtering, the same Seurat-based pre-processing pipeline is used as follows: raw counts containing only expressed genes were normalised using the 'NormalizeData' function and the top 2000 highly variable genes were selected using the 'FindVariableFeatures' function which were subsequently scaled using the 'ScaleData' function. The top 30 PCs were calculated using 'RunPCA' function, which were then subjected to Harmony (v0.1.1) integration (*Korsunsky et al., 2019*) to remove patient-related batch effect within each dataset. The harmonised Harmony embeddings were used for Uniform Manifold Approximation and Projection (UMAP) visualisation and unsupervised clustering. Specifically, the 'FindNeighbors' function were ran on the 'harmony' reduction and 'FindClusters' function with a resolution of 1.0 were used. Clusters that express both PTPRC+ CD68+ (putative myeloid cells) were retained in the first round of cell filtering. These putative myeloid cells were re-normalised, harmonised, and clustered using the same pre-processing pipeline described above. Here, cell clusters expressing NK cell markers (*GZMB, GNLY, CCR7*) (*Mulder et al., 2021*), dendritic cell markers (*CD1C, FCER1A*) (*Mulder et al., 2021*), proliferation markers (*STMN1, TUBB*) (*Mulder et al., 2021*), and mesenchymal markers (*DCN, LUM*) (*Muhl et al., 2020*) were filtered out to derive a final set of monocyte/macrophages across 15 studies and 6 tissues used in this study.

### Integration and clustering of monocyte/macrophages at the tissue level

In order to harmonise the monocyte/macrophage single-cell data at the tissue level, the filtered monocyte/macrophages from each tissue were combined and subjected to data integration as follows: for each dataset, we retain genes that are expressed in two or more datasets within the tissue. SCTransform (v0.3.5) normalisation (*Hafemeister and Satija, 2019*) is then performed on each dataset separately. In order to correct for intra-dataset batch effects, regression for patient-related batch effect

(i.e., the patient covariate) was applied. All datasets are combined and subjected to Seurat CCA integration using the top 3000 highly variable genes to remove batch effects in-between datasets. This resulted in 10,401, 184,297, 24,461, 2776, 6030, and 7965 monocyte/macrophages from the liver, lung, heart, skin, endometrium, and kidney, respectively.

To derive clusters within the monocyte/macrophage population in an unbiased manner, unsupervised cluster analysis was performed using the 'FindNeighbors' and 'FindClusters' functions. Specifically, the analysis was performed multiple times with different cluster resolutions (from 0.20 to 1.00 in increments of.05), and the cluster output at each resolution was assessed by the Silhouette score (*Leary et al., 2023*). Briefly, the Silhouette score evaluates the separability of clustering by measuring for every cell the similarity with cells in the same cluster as compared to the similarity to cells in other clusters. We consider a cluster resolution lower than 0.3 as 'under-clustering', and for each tissue, we selected a resolution ≥0.3 that optimises the Silhouette score (*Figure 1—figure supplements 1–6A*). We noted that this data-driven approach may produce numerous clusters that share little transcriptomic differences. To merge clusters with little transcriptional differences, we constructed a pseudo-bulk expression profile of each cluster and assessed transcriptional similarity between every cluster pair using the Pearson correlation of the pseudo-bulk expression profiles of the top 30 PCs in the dataset (*Figure 1—figure supplements 1–6B*). Cluster pairs with Pearson correlation >0.6 were merged (*Figure 1—figure supplements 1–6C*). Marker genes for the merged clusters were calculated by Seurat's 'FindAllMarkers' function. Each cluster was annotated by marker genes in reference to known markers of homeostatic monocyte and macrophages, as well as SPP1$^+$ macrophages (see *Figure 1—source data 1 and 2*; *Figure 1—figure supplements 1–6C*). We plotted the top marker genes, with average log$_2$-fold-change (log$_2$FC)>3, for each cluster in a heatmap in *Figure 1—figure supplements 1–6D*, and highlighted the above-mentioned known markers.

## Identification of the MAM polarisation state

We applied the same Seurat integration pipeline (as mentioned above) to integrate the SPP1$^+$ macrophages from all tissues (liver, lung, heart, skin, endometrium, and kidney). Briefly, SCTransform normalisation regressing for patient covariate is performed to remove intra-dataset batch effects while Seurat CCA integration is performed to correct for inter-dataset technical bias. This resulted in pooling 359, 16,685, 3593, 123, 1626, and 1719 SPP1$^+$ macrophages from the liver, lung, heart, skin, endometrium, and kidney, respectively. The integrated gene expression matrix was scaled, subjected to PCA, unsupervised clustering and the optimal clustering resolution is determined as described above (*Figure 2—figure supplement 1A–C*). Nine clusters are identified at a resolution of 0.30 (*Figure 2—figure supplement 1D*). Marker genes for these clusters are calculated using Seurat's 'FindAllMarkers' function and one cluster showed a significantly higher expression of SPP1 among the SPP1$^+$ macrophages, which we refer to as SPP1$^+$MAM$^+$ macrophages throughout the manuscript (*Figure 2—figure supplement 1E*, *Figure 2—source data 4*). The remaining SPP1$^+$ macrophages are denoted as SPP1$^+$MAM$^-$ macrophages throughout the manuscript.

## SPP1$^+$MAM$^+$ and SPP1$^+$MAM$^-$ transcriptomic signatures

Differential expression is performed between SPP1$^+$MAM$^+$ and SPP1$^+$MAM$^-$ macrophages to identify genes that are associated with the MAM polarisation state. First, for each gene, the log$_2$FC between SPP1$^+$MAM$^+$ and SPP1$^+$MAM$^-$ macrophages is calculated using Seurat's 'FoldChange' function, which was run separately for each tissue. The log$_2$FCs were then averaged across tissues. Second, we calculated for each gene the specificity of expression using the difference in the proportion of cells (diff.pct) expressing the gene between SPP1$^+$MAM$^+$ and SPP1$^+$MAM$^-$ macrophages. The diff.pct were averaged across tissues and a one-tailed z-test was then performed to determine the statistically significant difference between cell proportions. Genes with log$_2$FC >0.5 and zscore(diff.pct)>1.96 i.e., genes that are specifically upregulated in SPP1$^+$MAM$^+$ are SPP1$^+$MAM$^+$ signature genes; n=158 genes (*Figure 2—source data 5*). Similarly, an SPP1$^+$MAM$^-$ transcriptomic signature is determined using cutoffs of log$_2$FC <–0.5 and scaled(diff.pct)<–1.96 (*Figure 5—source data 1*). We also performed GSEA using the ClusterProfiler package (v 4.6.0) on the log$_2$FC between SPP1$^+$MAM$^+$ and SPP1$^+$MAM$^-$ macrophages, querying the MSigDB Human C2 annotated pathways gene set database (*Liberzon et al., 2015*; *Figure 2D* and *Figure 2—figure supplement 1F*). We found that ECM remodelling and metabolism-related pathways are upregulated and the leading

edge genes (genes contributing to the gene set enrichment) within the top 53 terms are prioritised (using a cutoff of NES >1.6 and p.adjust <0.05). These leading edge genes are then overlapped with the SPP1+MAM+ gene signature, resulting in SPP1+MAM+ ECM remodelling (n=18 genes) and SPP1+MAM+ metabolic processes gene sets (n=27 genes), respectively (*Figure 2—source data 4 and 5*).

## SPP1+MAM+ marker genes

COMET (v 0.1.13) was used to derive marker genes of SPP1+MAM+ macrophages (*Delaney et al., 2019*). COMET is a marker panel selection tool for single-cell data, which employs XL-minimal hypergeometric test to binarise the expression of genes in order to comparatively assess whether they can be representative of a given cell cluster. The SCTransform-normalized expression matrix, comprising 10,514 genes, of the SPP1+ macrophages (SPP1+MAM+ and SPP1+MAM- cells) and the MAM+/MAM- assignment were used as input for COMET, which was run to identify four-marker panels. We noted that the same genes occurred multiple times in the top 2000 four-marker combinations predicted by COMET, which showed p-value $<10^{-300}$ (smaller than python floating point precision). We therefore considered all four-marker panel predictions and ranked each gene based on the number of occurrences in the predictions and selected the top 7 genes as candidate ones that distinguish SPP1+MAM+ from SPP1+MAM- macrophages using the Elbow method (*Figure 2—figure supplement 2*). Briefly, on a scatterplot we ranked each marker gene from COMET based on the number of occurrences, with the x-axis denoting the rank and the y-axis the number of occurrences. A line has been drawn between the dots representing the top ranked gene to the least ranked gene. We then calculated the shortest distance of each dot (which represents a marker gene) to the line. We identified the marker gene that maximizes this distance, and retained it together with other genes ranked higher as markers of SPP1+MAM+ macrophages. We further overlapped the top 7 genes with the SPP1+MAM+ gene signature to obtain a final refined set of six marker genes representative of SPP1+MAM+ macrophages. The expression of these genes was overlaid on a UMAP plot as shown in *Figure 2B*.

## Comparison of SPP1+MAM+ transcriptomic signatures with TAMs, LAMs, and DAM

Expression level of SPP1+MAM+ ECM remodelling and metabolic signatures were compared to those that define TAMs, LAMs, and DAM. Expression data of TAMs from *Mulder et al., 2021*, was obtained from *FG Lab, 2023*; LAMs from Jaitin et al. (GEO: GSE128518) (*Jaitin et al., 2019*) and Wirka et al. (GEO: GSE131780) (*Wirka et al., 2019*); DAM from Kumar et al. obtained from https://epicimmuneatlas.org/NatNeu2022/ (*Kumar et al., 2022b*). TREM2+ TAMs ('TREM2 Macrophage -3' cluster identified in *Mulder et al., 2021*) from lung, liver, skin, and colon were extracted based on single-cell meta information provided. The adipose tissue LAMs from *Jaitin et al., 2019*, were extracted based on the provided cell-type annotation. To extract expression data from coronary atherosclerotic plaque LAMs, raw expression data from Wirka et al. was processed following the authors' methods as closely as possible, using the parameters provided in the original publication (*Wirka et al., 2019*). DAM were defined as microglia clusters isolated from temporal lobe epilepsy patients provided by *Kumar et al., 2022b*, and only temporal lobe microglia samples processed within the same batch were kept ('P6.A', 'P4', 'P5.A', P3.A'). The microglia were then scored for transcriptomic signature of DAM using Seurat's AddModuleScore function using default parameters (see *Figure 2—source data 2* for marker genes of DAM signature), and a z-score for DAM signature score was calculated. TREM2+APOE+SPP1+ microglia clusters with the highest median signature score of DAM (top 25th percentile of z-score;DAM score = 0.376) were retained as DAM for further analysis (*Figure 2—figure supplement 3*). In order to compare expression levels in tissue macrophages derived from different tissues, SPP1+ macrophages, TAMs, LAMs, and DAM were integrated using the same Seurat integration pipeline as mentioned above. Briefly, SCTransform normalisation regressing for patient covariate is performed to remove intra-dataset batch effects while Seurat CCA integration is performed to correct for inter-dataset technical differences. To evaluate the strength of the SPP1+MAM+ ECM remodelling and metabolic signatures, Seurat's 'AddModuleScore' function was applied to the SCT assay of the final integrated dataset.

## Trajectory and cell differentiation propensity analyses

### Differentiation trajectory analysis

The R package Slingshot (v 2.4.0) was used to perform differentiation trajectory analysis on macrophages, separately in each tissue (*Street et al., 2018*). Briefly, Slingshot derives differentiation paths from a specified origin and calculates for each cell a pseudotime, which approximates differentiation progression of a given cell towards the destination of the trajectory. The UMAP dimensionality reduction and cell-type clusters (e.g., FCN1$^+$ monocyte, SPP1$^+$ macrophages, etc.) were used as input, and FCN1$^+$ monocyte cluster was specified as the origin of differentiation. One trajectory leading to SPP1$^+$MAM$^+$ was consistently recapitulated across different tissues. Slingshot curves were constructed using the 'slingCurves' function and pruned to accurately portray the trajectories. We further applied Monocle (v 3.16) (*Cao et al., 2019*) to validate the differentiation trajectories predicted by Slingshot. The Seurat objects used earlier were first converted to CellDataSet (CDS) objects using the Seurat-Wrappers library (v 0.3.1). Cluster information and UMAP coordinates were passed to the CDS object and a graph was constructed using the UMAP dimensionality reduction.

### Differentiation propensity analysis

We further extracted SPP1$^+$MAM$^+$, SPP1$^+$MAM$^-$, and transitional macrophages to examine the propensity of an SPP1$^+$MAM$^-$ macrophage to differentiate into SPP1$^+$MAM$^+$ macrophage in disease or control states. Here, we define transitional macrophages as cells that are not homeostatic/RNASE1$^+$ and SPP1$^+$. Diffusion map was employed to derive the probability of a given cell to differentiate into SPP1$^+$MAM$^+$. The diffusion map is a dimensionality reduction algorithm that recovers distance measure between a pair of cells with respect to the transitional probability from one cell to another based on random walk. We approximated the differentiation potential of a cell using transitional probability. Seurat objects were converted to SingleCellExperiment objects retaining the integrated assay. The R package Destiny (v 3.10.0) was used to build a diffusion map on transitional macrophages, SPP1$^+$MAM$^-$ macrophages and SPP1$^+$MAM$^+$ separately for each tissue (*Haghverdi et al., 2015*). The data objects were converted into SingleCellExperiment objects prior to calculating the diffusion map, which used the top 50 PCs and number of nearest neighbours (k) as 10% of the number of cells in the object. For the lung, the dataset was down-sampled to 25,000 cells before calculating diffusion map. The expression data of the top 2000 highly variable genes was used as input. The propensity of SPP1$^+$MAM$^-$ macrophages to differentiate into SPP1$^+$MAM$^+$ was approximated as the quotient of the probability of a cell to transition into SPP1$^+$MAM$^+$ over the sum of the probability of SPP1$^+$MAM$^-$ to transition into SPP1$^+$MAM$^+$ or a transitional macrophage. Since we focus on assessing probability of cell transitions, we are not considering the probability of cells retaining the same state, where P(SPP1$^+$MAM$^+$|SPP1$^+$MAM$^-$) denotes the probability of an SPP1$^+$MAM$^-$ macrophage to transition into SPP1$^+$MAM$^+$, and P(TransMac|SPP1$^+$MAM$^-$) denotes the probability of an SPP1$^+$MAM$^-$ macrophage to transition back to transitional macrophage. The calculation was performed separately for each disease group for each tissue.

$$\text{Propensity Score} = \frac{\sum \text{P}(\text{SPP1}^+\text{MAM}^+ \mid \text{SPP1}^+\text{MAM}^-)}{\sum \text{P}(\text{SPP1}^+\text{MAM}^+ \mid \text{SPP1}^+\text{MAM}^-) + \sum \text{P}(\text{Trans Mac} \mid \text{SPP1}^+\text{MAM}^-)}$$

## Regulon analysis

Regulon (gene-regulatory network) analysis was performed using pySCENIC (v 0.12.0) to derive a set of regulons likely driving the differentiation from SPP1$^+$MAM$^-$ to SPP1$^+$MAM$^+$ macrophages (*Van de Sande et al., 2020*). Briefly, pySCENIC (1) derives a set of gene co-expression networks defined by a TF and its target genes, (2) evaluates a network for enrichment of TF-specific cis-regulatory elements and removes target genes lacking an enrichment for those elements, and (3) assesses the activity of the network in each individual cell by an 'area under the curve' (AUC) score. We ran pySCENIC with default parameterisation on SPP1$^+$MAM$^-$ and SPP1$^+$MAM$^+$ macrophages. The expression data from integrated assay (output of Seurat's integration pipeline) (24,105 cells by 3000 genes) were used as input, and a list of human-specific TFs was downloaded from GitHub (*Van de Sande et al., 2022*). For each gene in the transcriptome, a tree-based regression model was built with the TF candidates as predictors using GRNBoost2. In step 2 (network refinement), we used the following database of genome-wide regulatory features (hg19-500bp-upstream-10species.mc9nr.genes_vs_motifs.rankings.

feather, hg19-tss-centered-5kb-10species.mc9nr.genes_vs_motifs.rankings.feather) and TF motifs (motifs-v9-nr.hgnc-m0.001-o0.0.tbl) provided by the laboratory of Serin Aerts, to assess a regulon for the enrichment of regulatory features and to prune the genes in the regulon. Briefly, these database files contain pre-computed rankings of genome-wide regulatory features in target genes. In step 3 (the evaluation of regulon activity), pySCENIC ranked each gene in the transcriptome of a cell by its expression. Based on this ranking, an AUC score evaluates the enrichment of the regulon genes. Finally, the activation status of a regulon in a cell is derived by binarising the AUC score.

In total, 238 regulons were identified, and we retained regulons which are activated in at least 10% of the cells in at least four tissues, resulting in 173 regulons. We evaluated the specificity of a regulon to SPP1$^+$MAM$^+$ macrophages using three criteria: (1) the specificity to the activation status of the regulon with respect to SPP1$^+$MAM$^+$ macrophages (relative to SPP1$^+$MAM$^-$ macrophages), (2) the specificity of the regulon to the core signature of SPP1$^+$MAM$^+$ (see SPP1$^+$MAM$^+$ transcriptome signature for details), and (3) the potential of the regulon to promote polarisation of SPP1$^+$MAM$^-$ macrophages toward SPP1$^+$MAM$^+$ state.

*First criterion* – we evaluated the specificity of the activation of a regulon in SPP1$^+$MAM$^+$ cells, using the diagnostic odd ratio (DOR) (*Angelidis et al., 2019*). DOR assesses the odds of a positive test in 'cases' relative to the odds of a positive test in 'controls'. Here, we refer to the activation of the regulon in SPP1$^+$MAM$^+$ as 'cases' and the activation of the regulon in SPP1$^+$MAM$^-$ macrophage as 'controls'. The DOR for a regulon is calculated using the following formula:

$$\text{DOR} = \frac{(\text{TP} + 0.05)\,(\text{TN} + 0.05)}{(\text{FP} + 0.05)\,(\text{FN} + 0.05)}$$

where TP refers to number of SPP1$^+$MAM$^+$ in which the regulon is activated; TN to the number of SPP1$^+$MAM$^-$ macrophages in which the regulon is not activated; FP to the number of SPP1$^+$MAM$^-$ macrophages in which the regulon is activated; FN to the number of SPP1$^+$MAM$^+$ macrophages in which the regulon is not activated. The DOR for each regulon is calculated for each tissue and the median DOR across six tissues is taken as the final score of regulon activation in SPP1$^+$MAM$^+$ macrophages.

*Second criterion* – to assess the specificity of a regulon with regard to the core signature of SPP1$^+$MAM$^+$ macrophages, we examined (1) the size of overlap between the regulon and the signature, and (2) the importance of the genes in the overlap. Specifically, we first performed a hypergeometric test to evaluate the significance of the overlap between the core SPP1$^+$MAM$^+$ signature and genes of the regulon. We then calculated and summed over the gene expression fold changes between the SPP1$^+$MAM$^+$ to SPP1$^+$MAM$^-$ macrophage states of the overlapping genes. We repeated the same calculation (hypergeometric test and fold-change calculation) for genes downregulated in SPP1$^+$MAM$^+$ compared to SPP1$^+$MAM$^-$ (log$_2$FC $<-0.5$). The specificity of a regulon to the core SPP1$^+$MAM$^+$ signature was then calculated using the following formula:

$$\text{Specificity} = \frac{[-\log(\text{p} - \text{val}_{\text{SPP1+MAM+}}) - (-\log(\text{p} - \text{val}_{\text{SPP1+MAM-}})] * (\sum \log_2 \text{FC}_{\text{SPP1+MAM+}} - \sum \log_2 \text{FC}_{\text{SPP1+MAM-}})}{\sqrt{\text{Size}_{\text{Regulon}}}}$$

where 'p-val' refers to the p-value of the hypergeometric test, and Size$_{\text{Regulon}}$ to the number of genes in the regulon. The expression of a regulon is calculated using AddModuleScore() function in *Seurat* R package.

*Third criterion* – to delineate which regulon is more specifically required for the polarisation of SPP1$^+$MAM$^-$ macrophages into the SPP1$^+$MAM$^+$ state (as opposed to the possible differentiation from transitional to SPP1$^+$MAM$^-$ macrophages), we pooled together SPP1$^+$MAM$^+$, transitional and SPP1$^+$MAM$^-$ macrophages from all tissues and plotted the activity score of a regulon in each macrophage against the activity score of the SPP1$^+$MAM$^+$ signature (*Figure 4C*) for selected regulons (see *Figure 4—source data 1* for genes of each regulon in this analysis). Activity score AUC is calculated using *AUCell* R package, which is the same as the one used in the final step of pySCENIC. Here, we evaluated the potential of SPP1$^+$MAM$^+$ macrophages to acquire the SPP1$^+$MAM$^+$ polarisation state using the expression of SPP1$^+$MAM$^+$ signature. We performed linear regression analyses of regulon expression against expression of the SPP1$^+$MAM$^+$ signature, using the transitional and SPP1$^+$MAM$^-$ macrophages, or using SPP1$^+$MAM$^-$ and SPP1$^+$MAM$^+$ macrophages. This allows us to identify any potential regulon that has a greater increase in regulon expression within the SPP1$^+$MAM$^-$ and

SPP1+MAM+ macrophages (as compared to SPP1+MAM- and transitional macrophages) as these regulons are most likely to drive the differentiation progression from SPP1+MAM- macrophages to the SPP1+MAM+ polarisation state.

## Evaluation of SPP1+MAM+ and SPP1+MAM- macrophage signatures during ageing

### Association of SPP1+MAM+ and SPP1+MAM- signatures with ageing in humans

We extracted human lung control monocytes and macrophages from the Human Lung Cell Atlas Core dataset (n=94) (*Sikkema et al., 2023*). Control monocytes and macrophages were retained using the provided cell identities to create a new object which was processed further. Briefly, the cell counts were normalized and scaled using the top 2000 variable features. The top 50 PCs were computed and used to find nearest neighbours, and the clustering was performed at a resolution of 0.5. Classical and non-classical monocyte clusters were identified by differential gene expression of monocyte markers (*FCN1*, *LST1*, *S100A8*, *S100A9*, and *FCGR3A*). In order to create a human-specific homeostatic macrophage signature, differentially expressed genes were identified between RNASE1+ macrophages and all other macrophages in each tissue. The difference in proportion of cells expressing each gene and the average $\log_2$FC were averaged across all tissues. Differentially expressed genes having a $\log_2$FC >0.5, an averaged z-scaled difference in proportion >1.96 (corresponding to 0.05 for the z-test), and an FDR <0.05 were selected for the homeostatic macrophage gene signature. All cells were scored for the homeostatic RNASE1+, SPP1+MAM-, SPP1+MAM+, SPP1+MAM+ ECM remodelling and SPP1+MAM+ metabolic signatures using Seurat's AddModuleScore function. Donors with less than 10 cells were excluded from the analysis. Monocytes were then filtered out and a macrophage-only object (n=85, 86, 484 cells) was used to identify an association with ageing. Pseudobulk scores were then constructed per-patient by taking the median of the scores, for each signature. We used a linear regression model where each signature score was regressed against the patient's age, sex, and smoking status.

### Association of SPP1+MAM+ and SPP1+MAM- signatures with ageing in mice

We used the Tabula Muris Senis dataset for this analysis (*Kang et al., 2015*). The provided cell identities were used to subset the monocytes and macrophages. We retained cells from young (1–3 months) or old (21–30 months) mice. We performed the analysis on the lung and the kidney since other tissues had few monocytes/macrophages per animal. Classical and non-classical monocytes were identified by differential gene expression of monocyte markers (*Fcn1*, *Lst1*, *S100a8*, *S100a9*, *Fcgr3a*), as per the human dataset. The human signatures were converted to their murine orthologs using the R package babelgene (v 22.9). Each murine cell was then scored for the SPP1+MAM-, SPP1+MAM+, SPP1+MAM+ ECM remodelling and SPP1+MAM+ metabolic signatures using Seurat's AddModuleScore function. The monocytes were filtered out and the mice were grouped according to age (young vs. aged), prior to plotting.

## Acknowledgements

The research was primarily supported by the Academic Research Council, Ministry of Education (ARC, MOE) for AcRF Tier 2 funding (T2EP30221-0013) to EP; AcRF Tier 1 (2022-MOET1-0003) to JB. JB acknowledges Intramural Goh Cardiovascular Research Award from Duke NUS (Duke-NUS-GCR/2022/0020). This research is supported by the Singapore Ministry of Health's National Medical Research Council under its Open Fund Large Collaborative Grant (OFLCG22may-0011) and from Duke-NUS Medical School (EP and JB). JFO is supported by the Singapore National Medical Research Council (NMRC) under OF-YIRG funding (MOH-OFYIRG21nov-0004).

## Additional information

### Funding

| Funder | Grant reference number | Author |
|---|---|---|
| Ministry of Education - Singapore | T2EP30221-0013 | Enrico Petretto |
| Ministry of Education - Singapore | 2022-MOET1-0003 | Jacques Behmoaras |
| Duke-NUS | Intramural Goh Cardiovascular Research Award (Duke-NUS-GCR/2022/0020) | Jacques Behmoaras |
| National Medical Research Council | OFLCG22may-0011 | Enrico Petretto Jacques Behmoaras |
| Duke-NUS Medical School | | Enrico Petretto Jacques Behmoaras |
| National Medical Research Council | MOH-OFYIRG21nov-0004 | John F Ouyang |

The funders had no role in study design, data collection and interpretation, or the decision to submit the work for publication.

### Author contributions

John F Ouyang, Conceptualization, Data curation, Formal analysis, Supervision, Validation, Investigation, Methodology, Writing - review and editing; Kunal Mishra, Data curation, Formal analysis, Investigation, Visualization, Writing - review and editing; Yi Xie, Harry Park, Data curation, Formal analysis; Kevin Y Huang, Conceptualization, Methodology; Enrico Petretto, Conceptualization, Data curation, Supervision, Funding acquisition, Validation, Investigation, Methodology, Writing - original draft, Project administration; Jacques Behmoaras, Conceptualization, Data curation, Supervision, Funding acquisition, Investigation, Methodology, Writing - original draft, Project administration, Writing - review and editing

### Author ORCIDs

John F Ouyang http://orcid.org/0000-0002-1239-1577
Kunal Mishra http://orcid.org/0000-0003-0460-2554
Kevin Y Huang http://orcid.org/0000-0002-2288-3620
Jacques Behmoaras http://orcid.org/0000-0002-5170-2606

### Decision letter and Author response

Decision letter https://doi.org/10.7554/eLife.85530.sa1
Author response https://doi.org/10.7554/eLife.85530.sa2

## Additional files

### Supplementary files
• MDAR checklist

### Data availability

Raw or pre-filtered UMI count matrices for the datasets used in this study were downloaded either from GEO repositories or Zenodo repositories or atlas websites or obtained directly from authors (see *Table 1* for more details). The processed Seurat object for each of the six tissues and SPP1 macrophages can be downloaded at Zenodo. All data were analysed with commonly used open-source software programs and packages as detailed in the Materials and methods section. The code is publicly available at GitHub (copy archived at *Ouyang, 2023*).

The following previously published datasets were used:

| Author(s) | Year | Dataset title | Dataset URL | Database and Identifier |
|---|---|---|---|---|
| Ramachandran P, Dobie R, Wilson-Kanamori JR, Dora EF, Henderson BEP, Luu NT, Portman JR, Matchett KP, Brice M, Marwick JA | 2019 | Resolving the fibrotic niche of human liver cirrhosis using single-cell transcriptomics | https://www.ncbi.nlm.nih.gov/geo/query/acc.cgi?acc=GSE136103 | NCBI Gene Expression Omnibus, GSE136103 |
| Adams TS, Schupp JC, Poli S, Ayaub EA, Neumark N, Ahangari F, Chu SG, Raby BA, Deluliis G, Januszyk M | 2020 | IPF Cell Atlas | https://www.ncbi.nlm.nih.gov/geo/query/acc.cgi?acc=GSE136831 | NCBI Gene Expression Omnibus, GSE136831 |
| Morse C, Tabib T, Sembrat J, Buschur KL, Bittar HT, Valenzi E, Jiang Y, Kass DJ, Gibson K, Chen W, Mora A, Benos PV, Rojas M, Lafyatis R | 2019 | Proliferating SPP1/MERTK-expressing macrophages in idiopathic pulmonary fibrosis | https://www.ncbi.nlm.nih.gov/geo/query/acc.cgi?acc=GSE128033 | NCBI Gene Expression Omnibus, GSE128033 |
| Reyfman PA, Walter JM, Joshi N, Anekalla KR, McQuattie-Pimentel AC, Chiu S, Fernandez R, Akbarpour M, Chen C-I, Ren Z, Verma R, Abdala-Valencia H, Nam K, Chi M, Han S, Gonzalez-Gonzalez FJ, Soberanes S, Watanabe S, Williams KJN, Flozak AS | 2018 | Single-Cell Transcriptomic Analysis of Human Lung Reveals Complex Multicellular Changes During Pulmonary Fibrosis II | https://www.ncbi.nlm.nih.gov/geo/query/acc.cgi?acc=GSE122960 | NCBI Gene Expression Omnibus, GSE122960 |
| Valenzi E, Bulik M, Tabib T, Morse C, Sembrat J, Trejo Bittar H, Rojas M, Lafyatis R | 2019 | Single-cell analysis reveals fibroblast heterogeneity and myofibroblasts in systemic sclerosis-associated interstitial lung disease | https://www.ncbi.nlm.nih.gov/geo/query/acc.cgi?acc=GSE128169 | NCBI Gene Expression Omnibus, GSE128169 |
| Koenig AL, Shchukina I, Amrute J, Andhey PS, Zaitsev K, Lai L, Bajpai G, Bredemeyer A, Smith G, Jones C, Terrebonne E, Rentschler SL, Artyomov MN, Lavine KJ | 2022 | Cellular Atlas of Human Heart Failure | https://www.ncbi.nlm.nih.gov/geo/query/acc.cgi?acc=GSE183852 | NCBI Gene Expression Omnibus, GSE183852 |
| Rao M, Wang X, Guo G, Wang L, Chen S, Yin P, Chen K, Chen L, Zhang Z, Chen X, Hu X, Hu S, Song J | 2021 | Single cell RNA sequencing of human failing heart | https://www.ncbi.nlm.nih.gov/geo/query/acc.cgi?acc=GSE145154 | NCBI Gene Expression Omnibus, GSE145154 |

*Continued on next page*

*Continued*

| Author(s) | Year | Dataset title | Dataset URL | Database and Identifier |
|---|---|---|---|---|
| Gur C, Wang S-Y, Sheban F, Zada M, Li B, Kharouf F, Peleg H, Aamar S, Yalin A, Braun-Moscovici Y, Jaitin DA, Meir-Salame T, Hagai E, Kragesteen BK, Avni B, Grisariu S, Bornstein C, Shlomi-Loubaton S, David E | 2022 | LGR5 expressing skin fibroblasts define a major hub perturbed in Systemic Sclerosis | https://www.ncbi.nlm.nih.gov/geo/query/acc.cgi?acc=GSE195452 | NCBI Gene Expression Omnibus, GSE195452 |
| Deng C-C, Y-F Hu, Zhu D-H, Cheng Q, J-J Gu, Feng Q-L, Zhang L-X, Y-P Xu, Wang D, Rong Z, Yang B | 2021 | Single-cell RNA-seq reveals fibroblast heterogeneity and increased mesenchymalfibroblastsin human skin fibrotic diseases | https://www.ncbi.nlm.nih.gov/geo/query/acc.cgi?acc=GSE163973 | NCBI Gene Expression Omnibus, GSE163973 |
| Tan et al | 2022 | Single cell analysis of endometriosis reveals a coordinated transcriptional program driving immunotolerance and angiogenesis across eutopic and ectopic tissues | https://www.ncbi.nlm.nih.gov/geo/query/acc.cgi?acc=GSE179640 | NCBI Gene Expression Omnibus, GSE179640 |
| Fonseca et al | 2022 | A single-cell transcriptomic analysis of endometriosis | https://www.ncbi.nlm.nih.gov/geo/query/acc.cgi?acc=GSE213216 | NCBI Gene Expression Omnibus, GSE213216 |
| Kuppe et al | 2020 | Decoding myofibroblast origins in human kidney fibrosis | https://doi.org/10.5281/zenodo.4059315 | Zenodo, 10.5281/zenodo.4059315 |
| Lake et al | 2021 | Aggregated, clustered single-cell RNA-seq data used in the KPMP Atlas Explorer v1.3 | https://www.kpmp.org/doi-collection/10-48698-92nk-e805 | Kidney Precision Medicine Project Atlas, 10.48698/92nk-e805 |
| Malone AF, Wu H, Fronick C, Fulton R, Gaut JP, Humphreys BD | 2020 | Single Cell Transcriptional Analysis of Donor and Recipient Immune Cell Chimerism in the Rejecting Kidney Transplant | https://www.ncbi.nlm.nih.gov/geo/query/acc.cgi?acc=GSE145927 | NCBI Gene Expression Omnibus, GSE145927 |
| Valenzi E, Bulik M, Tabib T, Morse C, Sembrat J, Trejo Bittar H, Rojas M, Lafyatis R | 2021 | Disparate interferon signaling and shared aberrant basaloid cells in single-cell profiling of idiopathic pulmonary fibrosis and systemic sclerosis-associated interstitial lung disease | https://www.ncbi.nlm.nih.gov/geo/query/acc.cgi?acc=GSE156310 | NCBI Gene Expression Omnibus, GSE156310 |

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
