## [Editor Report]

This important study deepens our understanding of macrophage phenotypes in pathological contexts and identifies a new macrophage state associated with tissue fibrosis, as well as putative drivers of this cellular state. The authors provide convincing evidence and performed a well-thought-out and thoroughly described computational analysis of single-cell RNA sequencing data. This work will be of broad interest to the fields of tissue inflammation, fibrosis, macrophage biology, and immunology.

---

## [Decision Letter]

**Decision letter after peer review:**

Thank you for submitting your article "Systems level identification of a matrisome-associated macrophage polarisation in multi-organ fibrosis" for consideration by *eLife*. Your article has been reviewed by 2 peer reviewers, and the evaluation has been overseen by a Reviewing Editor and David James as the Senior Editor. The following individuals involved in the review of your submission have agreed to reveal their identity: Stefan Bekiranov (Reviewer #1); Clement Cochain (Reviewer #2).

*Reviewer #2 (Recommendations for the authors):*

In the abstract and at the beginning of the discussion the authors refer to the SPP1+ macrophages as 'tissue resident', which is contrary to their own conclusions and to published experimental data, where in most contexts SPP1+ macrophages are monocyte-derived (e.g. liver PMID 32888418; heart PMID 35950218), this should be reworded as it is confusing.

The annotations of other macrophage subsets could be improved, and their biological identity given when it is clearly identifiable. For instance, in several organs, the 'RNASE1+' cluster seems to represent LYVE1+ tissue-resident macrophages (PMID 34995099; 30872492), and FABP4+ macrophages in the lung are likely tissue-resident alveolar macrophages?

In the abstract, the authors write that the SPP1+MAM+ state is "driven by NFATC1 and HIVEP3 regulons": for now, this remains a hypothesis as it is not tested experimentally, I would suggest rephrasing this sentence.

As this is a purely computational study, it would be better if the code used by the authors is fully shared.

Sharing the processed data (e.g. Seurat objects) for reanalysis/exploration by readers would also be desirable.

---

## [Author Response]

Reviewer #2 (Recommendations for the authors):In the abstract and at the beginning of the discussion the authors refer to the SPP1+ macrophages as 'tissue resident', which is contrary to their own conclusions and to published experimental data, where in most contexts SPP1+ macrophages are monocyte-derived (e.g. liver PMID 32888418; heart PMID 35950218), this should be reworded as it is confusing.

This is a very good point. The reviewer is correct and this has now been amended. We also note that PMID 35950218 was missing from our references; and this has now been added as an additional proof regrading the bone marrow origin of Spp1+ macrophages.

The annotations of other macrophage subsets could be improved, and their biological identity given when it is clearly identifiable. For instance, in several organs, the 'RNASE1+' cluster seems to represent LYVE1+ tissue-resident macrophages (PMID 34995099; 30872492), and FABP4+ macrophages in the lung are likely tissue-resident alveolar macrophages?

We thank the reviewer for this suggestion. In the revised manuscript, we have redefined the macrophage clusters with widely acknowledged markers such as LYVE1 and FABP4 (see revised Figure 1 and Supplementary Table 1). We also noted and report that not all the RNASE1+ macrophages are LYVE1+, suggesting that RNASE1 is potentially a more suitable marker for homeostatic macrophages. Regarding FABP4+ alveolar macrophages, we have now added an explanatory sentence in the revised manuscript.

In the abstract, the authors write that the SPP1+MAM+ state is "driven by NFATC1 and HIVEP3 regulons": for now, this remains a hypothesis as it is not tested experimentally, I would suggest rephrasing this sentence.

We agree. The regulon analysis has been updated given the revised MAM signature arising from the additional datasets (endometrium and liver). The NFATC1 and HIVEP regulons were still significant but others (JDP2. KLF3, CEBPD) gained in significance (revised Figure 4). Interestingly the association with ‘osteoclast regulon activity’ has gained significance given strong experimental evidence linking JDP2 to osteoclastogenesis [4, 5]. That said, we agree with the reviewer that any functional implication will require experimental validation – the regulon statement in the abstract has now been tuned-down in the revised version.

As this is a purely computational study, it would be better if the code used by the authors is fully shared.

Agreed. The code is publicly available at https://github.com/the-ouyang-lab/mam-reproducibility.

This has now been added to Methods.

Sharing the processed data (e.g. Seurat objects) for reanalysis/exploration by readers would also be desirable.

We now share all the processed data (Seurat objects) as part of our revised manuscript. The processed Seurat object for each of the six tissues and SPP1 macrophages can be downloaded at https://zenodo.org/record/8266711 (Also included in Methods).

References

1. Fred RG, Steen Pedersen J, Thompson JJ, Lee J, Timshel PN, Stender S, Opseth Rygg M, Gluud LL, Bjerregaard Kristiansen V, Bendtsen F *et al*: Single-cell transcriptome and cell type-specific molecular pathways of human non-alcoholic steatohepatitis. *Sci Rep* 2022, 12(1):13484.

2. Fonseca MAS, Haro M, Wright KN, Lin X, Abbasi F, Sun J, Hernandez L, Orr NL, Hong J, Choi-Kuaea Y *et al*: Single-cell transcriptomic analysis of endometriosis. *Nat Genet* 2023, 55(2):255-267.

3. Sikkema L, Ramirez-Suastegui C, Strobl DC, Gillett TE, Zappia L, Madissoon E, Markov NS, Zaragosi LE, Ji Y, Ansari M *et al*: An integrated cell atlas of the lung in health and disease. *Nat Med* 2023, 29(6):1563-1577.

4. Kawaida R, Ohtsuka T, Okutsu J, Takahashi T, Kadono Y, Oda H, Hikita A, Nakamura K, Tanaka S, Furukawa H: *Jun D*imerization protein 2 (JDP2), a member of the AP-1 family of transcription factor, mediates osteoclast differentiation induced by RANKL. *J Exp Med* 2003, 197(8):1029-1035.

5. Maruyama K, Fukasaka M, Vandenbon A, Saitoh T, Kawasaki T, Kondo T, Yokoyama KK, Kidoya H, Takakura N, Standley D *et al*: The transcription factor Jdp2 controls bone homeostasis and antibacterial immunity by regulating osteoclast and neutrophil differentiation. *Immunity* 2012, 37(6):1024-1036.